# Learning Shuffle Ideals
# Under Restricted Distributions

**Dongqu Chen**
Department of Computer Science
Yale University
dongqu.chen@yale.edu

## Abstract

The class of shuffle ideals is a fundamental sub-family of regular languages. The shuffle ideal generated by a string set $U$ is the collection of all strings containing some string $u \in U$ as a (not necessarily contiguous) subsequence. In spite of its apparent simplicity, the problem of learning a shuffle ideal from given data is known to be computationally intractable. In this paper, we study the PAC learnability of shuffle ideals and present positive results on this learning problem under element-wise independent and identical distributions and Markovian distributions in the statistical query model. A constrained generalization to learning shuffle ideals under product distributions is also provided. In the empirical direction, we propose a heuristic algorithm for learning shuffle ideals from given labeled strings under general unrestricted distributions. Experiments demonstrate the advantage for both efficiency and accuracy of our algorithm.

## 1 Introduction

The learnablity of regular languages is a classic topic in computational learning theory. The applications of this learning problem include natural language processing (speech recognition, morphological analysis), computational linguistics, robotics and control systems, computational biology (phylogeny, structural pattern recognition), data mining, time series and music ([7, 14–18, 20, 21]). Exploring the learnability of the family of formal languages is significant to both theoretical and applied realms.

Valiant's PAC learning model introduces a clean and elegant framework for mathematical analysis of machine learning and is one of the most widely-studied theoretical learning models ([22]). In the PAC learning model, unfortunately, the class of regular languages, or equivalently the concept class of deterministic finite automata (DFA), is known to be inherently unpredictable ([1, 9, 19]). In a modified version of Valiant's model which allows the learner to make membership queries, Angluin [2] has shown that the concept class of regular languages is PAC learnable.

Throughout this paper we study the *PAC learnability* of a subclass of regular languages, the class of *(extended) shuffle ideals*. The shuffle ideal generated by an augmented string $U$ is the collection of all strings containing some $u \in U$ as a (not necessarily contiguous) subsequence, where an *augmented string* is a finite concatenation of symbol sets (see Figure 1 for an illustration). The special class of shuffle ideals generated by a single string is called the *principal* shuffle ideals. Unfortunately, even such a simple class is not PAC learnable, unless RP=NP ([3]). However, in most application scenarios, the strings are drawn from some particular distribution we are interested in. Angluin et al. [3] prove under the uniform string distribution, principal shuffle ideals are PAC learnable. Nevertheless, the requirement of complete knowledge of the distribution, the dependence on the symmetry of the uniform distribution and the restriction of principal shuffle ideals lead to the lack of generality of the algorithm. Our main contribution in this paper is to present positive results

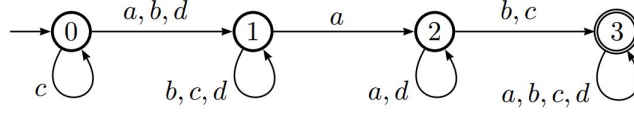

Figure 1: The DFA accepting precisely the shuffle ideal of $U = (a|b|d)a(b|c)$ over $\Sigma = \{a, b, c, d\}$.

on learning the class of shuffle ideals under element-wise independent and identical distributions and Markovian distributions. Extensions of our main results include a constrained generalization to learning shuffle ideals under product distributions and a heuristic method for learning principal shuffle ideals under general unrestricted distributions.

After introducing the preliminaries in Section 2, we present our main result in Section 3: the extended class of shuffle ideals is PAC learnable from element-wise i.i.d. strings. That is, the distributions of the symbols in a string are identical and independent of each other. A constrained generalization to learning shuffle ideals under product distributions is also provided. In Section 4, we further show the PAC learnability of principal shuffle ideals when the example strings drawn from $\Sigma^{\leq n}$ are generated by a Markov chain with some lower bound assumptions on the transition matrix. In Section 5, we propose a greedy algorithm for learning principal shuffle ideals under general unrestricted distributions. Experiments demonstrate the advantage for both efficiency and accuracy of our heuristic algorithm.

## 2  Preliminaries

We consider strings over a fixed finite alphabet $\Sigma$. The empty string is $\lambda$. Let $\Sigma^*$ be the Kleene star of $\Sigma$ and $\Sigma^{\cup}$ be the collection of all subsets of $\Sigma$. As strings are concatenations of symbols, we similarly define augmented strings as concatenations of unions of symbols.

**Definition 1 (Alphabet, simple string and augmented string)** *Let $\Sigma$ be a non-empty finite set of symbols, called the alphabet. A simple string over $\Sigma$ is any finite sequence of symbols from $\Sigma$, and $\Sigma^*$ is the collection of all simple strings. An augmented string over $\Sigma$ is any finite concatenation of symbol sets from $\Sigma^{\cup}$, and $(\Sigma^{\cup})^*$ is the collection of all augmented strings.*

Denote by $s$ the cardinality of $\Sigma$. Because an augmented string only contains strings of the same length, the length of an augmented string $U$, denoted by $|U|$, is the length of any $u \in U$. We use exponential notation for repeated concatenation of a string with itself, that is, $v^k$ is the concatenation of $k$ copies of string $v$. Starting from index 1, we denote by $v_i$ the $i$-th symbol in string $v$ and use notation $v[i, j] = v_i \ldots v_j$ for $1 \leq i \leq j \leq |v|$. Define the binary relation $\sqsubseteq$ on $\langle (\Sigma^{\cup})^*, \Sigma^* \rangle$ as follows. For a simple string $w$, $w \sqsubseteq v$ holds if and only if there is a witness $\vec{i} = (i_1 < i_2 < \ldots < i_{|w|})$ such that $v_{i_j} = w_j$ for all integers $1 \leq j \leq |w|$. For an augmented string $W$, $W \sqsubseteq v$ if and only if there exists some $w \in W$ such that $w \sqsubseteq v$. When there are several witnesses for $W \sqsubseteq v$, we may order them coordinate-wise, referring to the unique minimal element as the leftmost embedding. We will write $I_{W \sqsubseteq v}$ to denote the position of the last symbol of $W$ in its leftmost embedding in $v$ (if the latter exists; otherwise, $I_{W \sqsubseteq v} = \infty$).

**Definition 2 (Extended/Principal Shuffle Ideal)** *The (extended) shuffle ideal of an augmented string $U \in (\Sigma^{\cup})^L$ is a regular language defined as $\text{Ш}(U) = \{v \in \Sigma^* \mid \exists u \in U, u \sqsubseteq v\} = \Sigma^* U_1 \Sigma^* U_2 \Sigma^* \ldots \Sigma^* U_L \Sigma^*$. A shuffle ideal is principal if it is generated by a simple string.*

A shuffle ideal is an ideal in order theory and was originally defined for lattices. Denote by ⊔ the class of principal shuffle ideals and by Ш the class of extended shuffle ideals. Unless otherwise stated, in this paper shuffle ideal refers to the extended ideal. An example is given in Figure 1. The feasibility of determining whether a string is in the class $\text{Ш}(U)$ is obvious.

**Lemma 1** *Evaluating relation $U \sqsubseteq x$ and meanwhile determining $I_{U \sqsubseteq x}$ is feasible in time $O(|x|)$.*

In a computational learning model, an algorithm is usually given access to an oracle providing information about the sample. In Valiant's work [22], the example oracle $EX(c, \mathcal{D})$ was defined,

where $c$ is the target concept and $\mathcal{D}$ is a distribution over the instance space. On each call, $EX(c, \mathcal{D})$ draws an input $x$ independently at random from the instance space $\mathcal{I}$ under the distribution $\mathcal{D}$, and returns the labeled example $\langle x, c(x) \rangle$.

**Definition 3 (PAC Learnability: [22])** *Let $\mathcal{C}$ be a concept class over the instance space $\mathcal{I}$. We say $\mathcal{C}$ is probably approximately correctly (PAC) learnable if there exists an algorithm $\mathcal{A}$ with the following property: for every concept $c \in \mathcal{C}$, for every distribution $\mathcal{D}$ on $\mathcal{I}$, and for all $0 < \epsilon < 1/2$ and $0 < \delta < 1/2$, if $\mathcal{A}$ is given access to $EX(c, \mathcal{D})$ on $\mathcal{I}$ and inputs $\epsilon$ and $\delta$, then with probability at least $1 - \delta$, $\mathcal{A}$ outputs a hypothesis $h \in \mathcal{H}$ satisfying $\Pr_{x \in \mathcal{D}}[c(x) \neq h(x)] \leq \epsilon$. If $\mathcal{A}$ runs in time polynomial in $1/\epsilon$, $1/\delta$ and the representation size of $c$, we say that $\mathcal{C}$ is efficiently PAC learnable.*

We refer to $\epsilon$ as the error parameter and $\delta$ as the confidence parameter. If the error parameter is set to 0, the learning is exact ([6]). Kearns [11] extended Valiant's model and introduced the statistical query oracle $STAT(c, \mathcal{D})$. Kearns' oracle takes as input a statistical query of the form $(\chi, \tau)$. Here $\chi$ is any mapping of a labeled example to $\{0, 1\}$ and $\tau \in [0, 1]$ is called the noise tolerance. $STAT(c, \mathcal{D})$ returns an estimate for the expectation $\mathbf{E}\chi$, that is, the probability that $\chi = 1$ when the labeled example is drawn according to $\mathcal{D}$. A statistical query can have a condition so $\mathbf{E}\chi$ can be a conditional probability. This estimate is accurate within additive error $\tau$.

**Definition 4 (Legitimacy and Feasibility: [11])** *A statistical query $\chi$ is legitimate and feasible if and only if with respect to $1/\epsilon$, $1/\tau$ and representation size of $c$:*

1. *Query $\chi$ maps a labeled example $\langle x, c(x) \rangle$ to $\{0, 1\}$;*

2. *Query $\chi$ can be efficiently evaluated in polynomial time;*

3. *The condition of $\chi$, if any, can be efficiently evaluated in polynomial time;*

4. *The probability of the condition of $\chi$, if any, should be at least polynomially large.*

Throughout this paper, the learnability of shuffle ideals is studied in the statistical query model. Kearns [11] proves that oracle $STAT(c, \mathcal{D})$ is weaker than oracle $EX(c, \mathcal{D})$. In words, if a concept class is PAC learnable from $STAT(c, \mathcal{D})$, then it is PAC learnable from $EX(c, \mathcal{D})$, but not necessarily vice versa.

## 3 Learning shuffle ideals from element-wise i.i.d. strings

Although learning the class of shuffle ideals has been proved hard, in most scenarios the string distribution is restricted or even known. A very usual situation in practice is that we have some prior knowledge of the unknown distribution. One common example is the string distributions where each symbol in a string is generated independently and identically from an unknown distribution. It is element-wise i.i.d. because we view a string as a vector of symbols. This case is general enough to cover some popular distributions in applications such as the uniform distribution and the multinomial distribution. In this section, we present as our main result a statistical query algorithm for learning the concept class of extended shuffle ideals from element-wise i.i.d. strings and provide theoretical guarantees of its computational efficiency and accuracy in the statistical query model. The instance space is $\Sigma^n$. Denote by $U$ the augmented pattern string that generates the target shuffle ideal and by $L = |U|$ the length of $U$.

### 3.1 Statistical query algorithm

Before presenting the algorithm, we define function $\theta_{V,a}(\cdot)$ and query $\chi_{V,a}(\cdot, \cdot)$ for any augmented string $V \in (\Sigma^{\cup})^{\leq n}$ and any symbol $a \in \Sigma$ as as follows.

$$\theta_{V,a}(x) = \begin{cases} a & \text{if } V \not\sqsubseteq x[1, n-1] \\ x_{I_{V \sqsubseteq x}+1} & \text{if } V \sqsubseteq x[1, n-1] \end{cases}$$

$$\chi_{V,a}(x, y) = \frac{1}{2}(y+1) \quad \text{given } \theta_{V,a}(x) = a$$

where $y = c(x)$ is the label of example string $x$. More precisely, $y = +1$ if $x \in \text{Ш}(U)$ and $y = -1$ otherwise. Our learning algorithm uses statistical queries to recover string $U \in (\Sigma^\cup)^L$ one element at a time. It starts with the empty string $V = \lambda$. Having recovered $V = U[1, \ell]$ where $0 \le \ell < L$, we infer $U_{\ell+1}$ as follows. For each $a \in \Sigma$, the statistical query oracle is called with the query $\chi_{V,a}$ at the error tolerance $\tau$ claimed in Theorem 1. Our key technical observation is that the value of $\mathbf{E}\chi_{V,a}$ effectively selects $U_{\ell+1}$. The query results of $\chi_{V,a}$ will form two separate clusters such that the maximum difference (variance) inside one cluster is smaller than the minimum difference (gap) between the two clusters, making them distinguishable. The set of symbols in the cluster with larger query results is proved to be $U_{\ell+1}$. Notice that this statistical query only works for $0 \le \ell < L$. To complete the algorithm, the algorithm addresses the trivial case $\ell = L$ with query $\Pr[y = +1 \mid V \sqsubseteq x]$ and halts if the query answer is close to 1.

## 3.2 PAC learnability of ideal Ш

We show the algorithm described above learns the class of shuffle ideals from element-wise i.i.d. strings in the statistical query learning model.

**Theorem 1** *Under element-wise independent and identical distributions over instance space $\mathcal{I} = \Sigma^n$, concept class Ш is approximately identifiable with $O(sn)$ conditional statistical queries from STAT$(\text{Ш}, \mathcal{D})$ at tolerance*

$$\tau = \frac{\epsilon^2}{40sn^2 + 4\epsilon}$$

*or with $O(sn)$ statistical queries from STAT$(\text{Ш}, \mathcal{D})$ at tolerance*

$$\bar{\tau} = \left(1 - \frac{\epsilon}{20sn^2 + 2\epsilon}\right) \frac{\epsilon^4}{16sn(10sn^2 + \epsilon)}$$

We provide the main idea of the proofs in this section and defer the details and algebra to Appendix A. The proof starts from the legitimacy and feasibility of the algorithm. Since $\chi_{V,a}$ computes a binary mapping from labeled examples to $\{0, 1\}$, the legitimacy is trivial. But $\chi_{V,a}$ is not feasible for symbols in $\Sigma$ of small occurrence probabilities. We avoid the problematic cases by reducing the original learning problem to the same problem with a polynomial lower bound assumption $\Pr[x_i = a] \ge \epsilon/(2sn) - \epsilon^2/(20sn^2 + 2\epsilon)$ for any $a \in \Sigma$ and achieve feasibility.

The correctness of the algorithm is based on the intuition that the query result $\mathbf{E}\chi_{V,a_+}$ of a symbol $a_+ \in U_{\ell+1}$ should be greater than that of a symbol $a_- \notin U_{\ell+1}$ and the difference is large enough to tolerate the noise from the oracle. To prove this, we first consider the exact learning case. Define an infinite string $U' = U[1, \ell]U[\ell + 2, L]U_{\ell+1}^\infty$ and let $x' = x\Sigma^\infty$ be the extension of $x$ obtained by padding it on the right with an infinite string generated from the same distribution as $x$. Let $Q(j, i)$ be the probability that the largest $g$ such that $U'[1, g] \sqsubseteq x'[1, i]$ is $j$, or formally

$$Q(j, i) = \Pr[U'[1, j] \sqsubseteq x'[1, i] \wedge U'[1, j + 1] \not\sqsubseteq x'[1, i]]$$

By taking the difference between $\mathbf{E}\chi_{V,a_+}$ and $\mathbf{E}\chi_{V,a_-}$ in terms of $Q(j, i)$, we get the query tolerance for exact learning.

**Lemma 2** *Under element-wise independent and identical distributions over instance space $\mathcal{I} = \Sigma^n$, concept class Ш is exactly identifiable with $O(sn)$ conditional statistical queries from STAT$(\text{Ш}, \mathcal{D})$ at tolerance*

$$\tau' = \frac{1}{5}Q(L - 1, n - 1)$$

Lemma 2 indicates bounding the quantity $Q(L - 1, n - 1)$ is the key to the tolerance for PAC learning. Unfortunately, the distribution $\{Q(j, i)\}$ doesn't seem to have any strong properties we know of providing a polynomial lower bound. Instead we introduce new quantity

$$R(j, i) = \Pr[U'[1, j] \sqsubseteq x'[1, i] \wedge U'[1, j] \not\sqsubseteq x'[1, i - 1]]$$

being the probability that the smallest $g$ such that $U'[1, j] \sqsubseteq x'[1, g]$ is $i$. An important property of distribution $\{R(j, i)\}$ is its strong unimodality as defined below.

**Definition 5 (Unimodality: [8])** *A distribution $\{P(i)\}$ with all support on the lattice of integers is unimodal if and only if there exists at least one integer $K$ such that $P(i) \geq P(i-1)$ for all $i \leq K$ and $P(i+1) \leq P(i)$ for all $i \geq K$. We say $K$ is a mode of distribution $\{P(i)\}$.*

Throughout this paper, when referring to the mode of a distribution, we mean the one with the largest index, if the distribution has multiple modes with equal probabilities.

**Definition 6 (Strong Unimodality: [10])** *A distribution $\{H(i)\}$ is strongly unimodal if and only if the convolution of $\{H(i)\}$ with any unimodal distribution $\{P(i)\}$ is unimodal.*

Since a distribution with all mass at zero is unimodal, a strongly unimodal distribution is also unimodal. In this paper, we only consider distributions with all support on the lattice of integers. So the convolution of $\{H(i)\}$ and $\{P(i)\}$ is

$$\{H * P\}(i) = \sum_{j=-\infty}^{\infty} H(j)P(i-j) = \sum_{j=-\infty}^{\infty} H(i-j)P(j)$$

We prove the strong unimodality of $\{R(j,i)\}$ with respect to $i$ via showing it is the convolution of two log-concave distributions by induction. We do an initial statistical query to estimate $\Pr[y = +1]$ to handle two marginal cases $\Pr[y = +1] \leq \epsilon/2$ and $\Pr[y = +1] \geq 1 - \epsilon/2$. After that an additional query $\Pr[y = +1 \mid V \sqsubseteq x]$ is made to tell whether $\ell = L$. If the algorithm doesn't halt, it means $\ell < L$ and both $\Pr[y = +1]$ and $\Pr[y = -1]$ are at least $\epsilon/2 - 2\tau$. By upper bounding $\Pr[y = +1]$ and $\Pr[y = -1]$ using linear sums of $R(j,i)$, the strong unimodality of $\{R(j,i)\}$ gives a lower bound for $R(L,n)$, which further implies one for $Q(L-1, n-1)$ and completes the proof.

### 3.3 A generalization to instance space $\Sigma^{\leq n}$

We have proved the extended class of shuffle ideals is PAC learnable from element-wise i.i.d. fixed-length strings. Nevertheless, in many real-world applications such as natural language processing and computational linguistics, it is more natural to have strings of varying lengths. Let $n$ be the maximum length of the sample strings and as a consequence the instance space for learning is $\Sigma^{\leq n}$. Here we show how to generalize the statistical query algorithm in Section 3.1 to the more general instance space $\Sigma^{\leq n}$.

Let $\mathcal{A}_i$ be the algorithm in Section 3.1 for learning shuffle ideals from element-wise i.i.d. strings of fixed length $i$. Because instance space $\Sigma^{\leq n} = \bigcup_{i \leq n} \Sigma^i$, we divide the sample $S$ into $n$ subsets $\{S_i\}$ where $S_i = \{x \mid |x| = i\}$. An initial statistical query then is made to estimate probability $\Pr[|x| = i]$ for each $i \leq n$ at tolerance $\epsilon/(8n)$. We discard all subsets $S_i$ with query answer $\leq 3\epsilon/(8n)$ in the learning procedure, because we know $\Pr[|x| = i] \leq \epsilon/(2n)$. As there are at most $(n-1)$ such $S_i$ of low occurrence probabilities. The total probability that an instance comes from one of these negligible sets is at most $\epsilon/2$. Otherwise, $\Pr[|x| = i] \geq \epsilon/(4n)$ and we apply algorithm $\mathcal{A}_i$ on each $S_i$ with query answer $\geq 3\epsilon/(8n)$ with error parameter $\epsilon/2$. Because the probability of the condition is polynomially large, the algorithm is feasible. Finally, the total error over the whole instance space will be bounded by $\epsilon$ and concept class $\amalg$ is PAC learnable from element-wise i.i.d. strings over instance space $\Sigma^{\leq n}$.

**Corollary 1** *Under element-wise independent and identical distributions over instance space $\mathcal{I} = \Sigma^{\leq n}$, concept class $\amalg$ is approximately identifiable with $O(sn^2)$ conditional statistical queries from $STAT(\amalg, \mathcal{D})$ at tolerance*

$$\tau = \frac{\epsilon^2}{160sn^2 + 8\epsilon}$$

*or with $O(sn^2)$ statistical queries from $STAT(\amalg, \mathcal{D})$ at tolerance*

$$\bar{\tau} = \left(1 - \frac{\epsilon}{40sn^2 + 2\epsilon}\right) \frac{\epsilon^5}{512sn^2(20sn^2 + \epsilon)}$$

### 3.4 A constrained generalization to product distributions

A direct generalization from element-wise independent and identical distributions is product distributions. A random string, or a random vector of symbols under a product distribution has element-wise independence between its elements. That is, $\Pr[X = x] = \prod_{i=1}^{|x|} \Pr[X_i = x_i]$. Although strings under product distributions share many independence properties with element-wise i.i.d. strings, the algorithm in Section 3.1 is not directly applicable to this case as the distribution $\{R(j,i)\}$ defined above is not unimodal with respect to $i$ in general. However, the intuition that given $I_{V \sqsubseteq x} = h$, the strings with $x_{h+1} \in U_{\ell+1}$ have higher probability of positivity than that of the strings with $x_{h+1} \notin U_{\ell+1}$ is still true under product distributions. Thus we generalize query $\chi_{V,a}$ and define for any $V \in (\Sigma^{\cup})^{\leq n}$, $a \in \Sigma$ and $h \in [0, n-1]$,

$$\tilde{\chi}_{V,a,h}(x,y) = \frac{1}{2}(y+1) \quad \text{given } I_{V \sqsubseteq x} = h \text{ and } x_{h+1} = a$$

where $y = c(x)$ is the label of example string $x$. To ensure the legitimacy and feasibility of the algorithm, we have to attach a lower bound assumption that $\Pr[x_i = a] \geq t > 0$, for $\forall 1 \leq i \leq n$ and $\forall a \in \Sigma$. Appendix C provides a constrained algorithm based on this intuition. Let $P(+|a, h)$ denote $\mathbf{E}\tilde{\chi}_{V,a,h}$. If the difference $P(+|a_+, h) - P(+|a_-, h)$ is large enough for some $h$ with nonnegligible $\Pr[I_{V \sqsubseteq x} = h]$, then we are able to learn the next element in $U$. Otherwise, the difference is very small and we will show that there is an interval starting from index $(h+1)$ which we can skip with little risk. The algorithm is able to classify any string whose classification process skips $O(1)$ intervals. Details of this constrained generalization are deferred to Appendix C.

## 4 Learning principal shuffle ideals from Markovian strings

Markovian strings are widely studied in natural language processing and biological sequence modeling. Formally, a random string $x$ is Markovian if the distribution of $x_{i+1}$ only depends on the value of $x_i$: $\Pr[x_{i+1} \mid x_1 \dots x_i] = \Pr[x_{i+1} \mid x_i]$ for any $i \geq 1$. If we denote by $\pi_0$ the distribution of $x_1$ and define $s \times s$ stochastic matrix $M$ by $M(a_1, a_2) = \Pr[x_{i+1} = a_1 \mid x_i = a_2]$, then a random string can be viewed as a Markov chain with initial distribution $\pi_0$ and transition matrix $M$. We choose $\Sigma^{\leq n}$ as the instance space in this section and assume independence between the string length and the symbols in the string. We assume $\Pr[|x| = k] \geq t$ for all $1 \leq k \leq n$ and $\min\{M(\cdot, \cdot), \pi_0(\cdot)\} \geq c$ for some positive $t$ and $c$. We will prove the PAC learnability of class ⊔ under this lower bound assumption. Denote by $u$ be the target pattern string and let $L = |u|$.

### 4.1 Statistical query algorithm

Starting with empty string $v = \lambda$, the pattern string $u$ is recovered one symbol at a time. Having recovered $v = u[1, \ell]$, we infer $u_{\ell+1}$ by $\Psi_{v,a} = \sum_{k=h+1}^{n} \mathbf{E}\chi_{v,a,k}(x,y)$, where

$$\chi_{v,a,k}(x,y) = \frac{1}{2}(y+1) \quad \text{given } I_{v \sqsubseteq x} = h, \ x_{h+1} = a \text{ and } |x| = k$$

$0 \leq \ell < L$ and $h$ is chosen from $[0, n-1]$ such that the probability $\Pr[I_{v \sqsubseteq x} = h]$ is polynomially large. The statistical queries $\chi_{v,a,k}$ are made at tolerance $\tau$ claimed in Theorem 2 and the symbol with the largest query result of $\Psi_{v,a}$ is proved to be $u_{\ell+1}$. Again, the case where $\ell = L$ is addressed by query $\Pr[y = +1 \mid v \sqsubseteq x]$. The learning procedure is completed if the query result is close to 1.

### 4.2 PAC learnability of principal ideal ⊔

With query $\Psi_{v,a}$, we are able to recover the pattern string $u$ approximately from $STAT(\text{⊔}(u), \mathcal{D})$ at proper tolerance as stated in Theorem 2:

**Theorem 2** *Under Markovian string distributions over instance space $\mathcal{I} = \Sigma^{\leq n}$, given $\Pr[|x| = k] \geq t > 0$ for $\forall 1 \leq k \leq n$ and $\min\{M(\cdot, \cdot), \pi_0(\cdot)\} \geq c > 0$, concept class ⊔ is approximately identifiable with $O(sn^2)$ conditional statistical queries from $STAT(\text{⊔}, \mathcal{D})$ at tolerance*

$$\tau = \frac{\epsilon}{3n^2 + 2n + 2}$$

*or with $O(sn^2)$ statistical queries from STAT($\sqcup, \mathcal{D}$) at tolerance*

$$\bar{\tau} = \frac{3ctn\epsilon^2}{(3n^2 + 2n + 2)^2}$$

Please refer to Appendix B for a complete proof of Theorem 2. Due to the probability lower bound assumptions, the legitimacy and feasibility are obvious. To calculate the tolerance for PAC learning, we first consider the exact learning tolerance. Let $x'$ be an infinite string generated by the Markov chain defined above. For any $0 \leq \ell \leq L - j$, we define quantity $R_\ell(j, i)$ by

$$R_\ell(j, i) = \Pr[u[\ell+1, \ell+j] \sqsubseteq x'[m+1, m+i] \wedge u[\ell+1, \ell+j] \not\sqsubseteq x'[m+1, m+i-1] \mid x'_m = u_\ell]$$

Intuitively, $R_\ell(j, i)$ is the probability that the smallest $g$ such that $u[\ell+1, \ell+j] \sqsubseteq x'[m+1, m+g]$ is $i$, given $x'_m = u_\ell$. We have the following conclusion on the exact learning tolerance.

**Lemma 3** *Under Markovian string distributions over instance space $\mathcal{I} = \Sigma^{\leq n}$, given $\Pr[|x| = k] \geq t > 0$ for $\forall 1 \leq k \leq n$ and $\min\{M(\cdot, \cdot), \pi_0(\cdot)\} \geq c > 0$, the concept class $\sqcup$ is exactly identifiable with $O(sn^2)$ conditional statistical queries from STAT($\sqcup, \mathcal{D}$) at tolerance*

$$\tau' = \min_{0 \leq \ell < L} \left\{ \frac{1}{3(n-h)} \sum_{k=h+1}^{n} R_{\ell+1}(L-\ell-1, k-h-1) \right\}$$

The algorithm first deals with the marginal case where $P[y = +1] \leq \epsilon$ through query $\Pr[y = +1]$. If it doesn't halt, we know $\Pr[y = +1]$ is at least $(3n^2 + 2n)\epsilon/(3n^2 + 2n + 2)$. We then make a statistical query $\chi'_h(x, y) = \frac{1}{2}(y+1) \cdot \mathbb{1}_{\{I_{v \sqsubseteq x} = h\}}$ for each $h$ from $\ell$ to $n-1$. It can be shown that at least one $h$ will give an answer $\geq (3n+1)\epsilon/(3n^2 + 2n + 2)$. This implies lower bounds for $\Pr[I_{v \sqsubseteq x} = h]$ and $\Pr[y = +1 \mid I_{v \sqsubseteq x} = h]$. The former guarantees the feasibility while the latter can serve as a lower bound for the sum in Lemma 3 after some algebra and completes the proof.

The assumption on $M$ and $\pi_0$ can be weakened to $M(u_{\ell+1}, u_\ell) = \Pr[x_2 = u_{\ell+1} \mid x_1 = u_\ell] \geq c$ and $\pi_0(u_1) \geq c$ for all $1 \leq \ell \leq L - 1$. We first make a statistical query to estimate $M(a, u_\ell)$ for $\ell \geq 1$ or $\pi_0(a)$ for $\ell = 0$ for each symbol $a \in \Sigma$ at tolerance $c/3$. If the result is $\leq 2c/3$ then $M(a, u_\ell) \leq c$ or $\pi_0(a) \leq c$ and we won't consider symbol $a$ at this position. Otherwise, $M(a, u_\ell) \geq c/3$ or $\pi_0(a) \geq c/3$ and the queries in the algorithm are feasible.

**Corollary 2** *Under Markovian string distributions over instance space $\mathcal{I} = \Sigma^{\leq n}$, given $\Pr[|x| = k] \geq t > 0$ for $\forall 1 \leq k \leq n$, $\pi_0(u_1) \geq c$ and $M(u_{\ell+1}, u_\ell) \geq c > 0$ for $\forall 1 \leq \ell \leq L - 1$, concept class $\sqcup$ is approximately identifiable with $O(sn^2)$ conditional statistical queries from STAT($\sqcup, \mathcal{D}$) at tolerance*

$$\tau = \min \left\{ \frac{\epsilon}{3n^2 + 2n + 2}, \frac{c}{3} \right\}$$

*or with $O(sn^2)$ statistical queries from STAT($\sqcup, \mathcal{D}$) at tolerance*

$$\bar{\tau} = \min \left\{ \frac{ctn\epsilon^2}{(3n^2 + 2n + 2)^2}, \frac{tn\epsilon c^2}{3(3n^2 + 2n + 2)} \right\}$$

# 5  Learning shuffle ideals under general distributions

Although the string distribution is restricted or even known in most application scenarios, one might be interested in learning shuffle ideals under general unrestricted and unknown distributions without any prior knowledge. Unfortunately, under standard complexity assumptions, the answer is negative. Angluin et al. [3] have shown that a polynomial time PAC learning algorithm for principal shuffle ideals would imply the existence of polynomial time algorithms to break the RSA cryptosystem, factor Blum integers, and test quadratic residuosity.

**Theorem 3 ([3])** *For any alphabet of size at least 2, given two disjoint sets of strings $S, T \subset \Sigma^{\leq n}$, the problem of determining whether there exists a string $u$ such that $u \sqsubseteq x$ for each $x \in S$ and $u \not\sqsubseteq x$ for each $x \in T$ is NP-complete.*

As ideal ⊔ is a subclass of ideal Ⅲ, we know learning ideal Ⅲ is only harder. Is the problem easier over instance space $\Sigma^n$? The answer is again no.

**Lemma 4** *Under general unrestricted string distributions, a concept class is PAC learnable over instance space $\Sigma^{\leq n}$ if and only if it is PAC learnable over instance space $\Sigma^n$.*

The proof of Lemma 4 is presented in Appendix D using the same idea as our generalization in Section 3.3. Note that Lemma 4 holds under general string distributions. It is not necessarily true when we have assumptions on the marginal distribution of string length.

Despite the infeasibility of PAC learning a shuffle ideal in theory, it is worth exploring the possibilities to do the classification problem without theoretical guarantees, since most applications care more about the empirical performance than about theoretical results. For this purpose we propose a heuristic greedy algorithm for learning principal shuffle ideals based on reward strategy as follows. Upon having recovered $v = \widehat{u}[1, \ell]$, for a symbol $a \in \Sigma$ and a string $x$ of length $n$, we say $a$ consumes $k$ elements in $x$ if $\min\{I_{va \sqsubseteq x}, n+1\} - I_{v \sqsubseteq x} = k$. The reward strategy depends on the ratio $r_+/r_-$: the algorithm receives $r_-$ reward from each element it consumes in a negative example or $r_+$ penalty from each symbol it consumes in a positive string. A symbol is chosen as $\widehat{u}_{\ell+1}$ if it brings us most reward. The algorithm will halt once $\widehat{u}$ exhausts any positive example and makes a false negative error, which means we have gone too far. Finally the ideal $\sqcup\!\sqcup (\widehat{u}[1, \ell-1])$ is returned as the hypothesis. The performance of this greedy algorithm depends a great deal on the selection of parameter $r_+/r_-$. A clever choice is $r_+/r_- = \#(-)/\#(+)$, where $\#(+)$ is the number of positive examples $x$ such that $\widehat{u} \sqsubseteq x$ and $\#(-)$ is the number of negative examples $x$ such that $\widehat{u} \sqsubseteq x$. A more recommended but more complex strategy to determine the parameter $r_+/r_-$ in practice is cross validation.

A better studied approach to learning regular languages, especially the piecewise-testable ones, in recent works is kernel machines ([12, 13]). An obvious advantage of kernel machines over our greedy method is its broad applicability to general classification learning problems. Nevertheless, the time complexity of the kernel machine is $O(N^3 + n^2 N^2)$ on a training sample set of size $N$ ([5]), while our greedy method only takes $O(snN)$ time due to its great simplicity. Because $N$ is usually huge for the demand of accuracy, kernel machines suffer from low efficiency and long running time in practice. To make a comparison between the greedy method and kernel machines for empirical performance, we conducted a series of experiments on a real world dataset [4] with string length $n$ as a variable. The experiment results demonstrate the empirical advantage on both efficiency and accuracy of the greedy algorithm over the kernel method, in spite of its simplicity. As this is a theoretical paper, we defer the details on the experiments to Appendix D, including the experiment setup and figures of detailed experiment results.

## 6  Discussion

We have shown positive results for learning shuffle ideals in the statistical query model under element-wise independent and identical distributions and Markovian distributions, as well as a constrained generalization to product distributions. It is still open to explore the possibilities of learning shuffle ideals under less restricted distributions with weaker assumptions. Also a lot more work needs to be done on approximately learning shuffle ideals in applications with pragmatic approaches. In the negative direction, even a family of regular languages as simple as the shuffle ideals is not efficiently properly PAC learnable under general unrestricted distributions unless RP=NP. Thus, the search for a nontrivial properly PAC learnable family of regular languages continues. Another theoretical question that remains is how hard the problem of learning shuffle ideals is, or whether PAC learning a shuffle ideal is as hard as PAC learning a deterministic finite automaton.

## Acknowledgments

We give our sincere gratitude to Professor Dana Angluin of Yale University for valuable discussions and comments on the learning problem and the proofs. Our thanks are also due to Professor Joseph Chang of Yale University for suggesting supportive references on strong unimodality of probability distributions and to the anonymous reviewers for their helpful feedback.

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
