[Supplementary Material]

# Learning Shuffle Ideals
# Under Restricted Distributions
# (Appendix)

**Dongqu Chen**
Department of Computer Science
Yale University
dongqu.chen@yale.edu

## A   Proof of Theorem 1

We start our proof from showing the legitimacy and feasibility of the algorithm at the tolerance claimed in the theorem. We first provide a quick proof of Lemma 1.

**Lemma 1 (in the main paper)** *Evaluating relation $U \sqsubseteq x$ and meanwhile determining $I_{U \sqsubseteq x}$ is feasible in time $O(|x|)$.*

**Proof** The evalution can be done recursively. The base case is $U = \lambda$ , where $U \sqsubseteq x$ holds and $I_{U \sqsubseteq x} = 0$. If $U = U_1 U'$ where $U_1 \in \Sigma^{\cup}$, we search for the leftmost occurrence of $U_1$ in $x$. If there is no such occurrence, then $U \not\sqsubseteq x$ and $I_{U \sqsubseteq x} = \infty$. Otherwise, $x = y U_1 x'$, where $U_1 \not\sqsubseteq y$. Then $U \sqsubseteq x$ if and only if $U' \sqsubseteq x'$ and $I_{U \sqsubseteq x} = I_{U_1 \sqsubseteq x} + I_{U' \sqsubseteq x'}$. We continue recursively with $U'$ and $x'$. The total running time of this procedure is $O(|x|)$. ∎

**Lemma 5** *Under element-wise independent and identical distributions over instance space $\mathcal{I} = \Sigma^n$, the conditional statistical query $\chi_{V,a}$ is legitimate and feasible at tolerance*

$$\tau = \frac{\epsilon^2}{40sn^2 + 4\epsilon}$$

**Proof** First of all, the function $\chi_{V,a}$ computes a binary mapping from labeled examples $(x, y)$ to $\{0, 1\}$ and satisfies the definition of a statistical query. Given $\theta_{V,a}(x) = a$, that is, given $V \not\sqsubseteq x[1, n-1]$ or $x_{I_{V \sqsubseteq x}+1} = a$ if $V \sqsubseteq x[1, n-1]$, the query $\chi_{V,a}(x, y)$ returns 0 if $x$ is a negative example ($y = -1$) or returns 1 if $x$ is a positive example ($y = +1$).

From Lemma 1, evaluating the relation $V \sqsubseteq x$ and meanwhile determining $I_{V \sqsubseteq x}$ is feasible in time $O(n)$. Thus, $\theta_{V,a}(x)$ and then $\chi_{V,a}(x, y)$ can be efficiently evaluated.

For
$$\Pr[\theta_{V,a}(x) = a] = \Pr[V \not\sqsubseteq x[1, n-1]] +$$
$$\Pr[V \sqsubseteq x[1, n-1]] \cdot \Pr[x_{I_{V \sqsubseteq x}+1} = a \mid V \sqsubseteq x[1, n-1]]$$

in order to prove $\Pr[\theta_{V,a}(x) = a]$ not too small, we only need to show one of the two items in the sum is at least polynomially large.

We make an initial statistical query with tolerance $\tau = \epsilon^2/(40sn^2 + 4\epsilon)$ to estimate $\Pr[y = +1]$. If the answer is $\leq \epsilon - \tau$, then $\Pr[y = +1] \leq \epsilon$ and the algorithm outputs a hypothesis that all examples are negative. Otherwise, $\Pr[y = +1]$ is at least $\epsilon - 2\tau$, and the statistical query $\chi_{V,a}$ is used. As $V \sqsubseteq x[1, n-1] = U[1, \ell] \sqsubseteq x[1, n-1]$ is a necessary condition of $y = +1$, we have

$$\Pr[V \sqsubseteq x[1, n-1]] \geq \Pr[y = +1] \geq \epsilon - \frac{\epsilon^2}{20sn^2 + 2\epsilon}$$

Since $x_{I_{V \sqsubseteq x}+1}$ and $x[1, I_{V \sqsubseteq x}]$ are independent,

$$\Pr[x_{I_{V \sqsubseteq x}+1} = a \mid V \sqsubseteq x[1, n-1]] = \Pr[x_{I_{V \sqsubseteq x}+1} = a]$$

Because we don't have any knowledge of the distribution, we can't guarantee $\Pr[x_{I_{V \sqsubseteq x}+1} = a]$ is large enough for every $a \in \Sigma$. However, we notice that there is no need to consider symbols with small probabilities of occurrence. Now we show why and how. For each $a \in \Sigma$, execute a statistical query

$$\chi'_a(x, y) = \mathbb{1}_{\{x_i = a\}} \tag{1}$$

at tolerance $\tau$, where $\mathbb{1}_{\{\pi\}}$ represents the 0-1 truth value of the predicate $\pi$. Since the strings are element-wise i.i.d., the index $i$ can be any integer between 1 and $n$. If the answer from oracle *STAT* is $\leq \epsilon/(2sn) - \tau$, then $\Pr[x_i = a] \leq \epsilon/(2sn)$. For such an $a$, the probability that it shows up in a string is at most $\epsilon/(2s)$. Because there are at most $s - 1$ such symbols in $\Sigma$, the probability that any of them shows up in a string is at most $\epsilon/2$. Otherwise, $\Pr[x_i = a] \geq \epsilon/(2sn) - 2\tau$. Thus we only need to consider the symbols $a \in \Sigma$ such that $\Pr[x_i = a] \geq \epsilon/(2sn) - 2\tau$ and learn the ideal with error parameter $\epsilon/2$ so that the total error will be bounded within $\epsilon$. For algebra succinctness, we use a concise lower bound for $\Pr[x_i = a]$:

$$\Pr[x_i = a] \geq \frac{\epsilon}{2sn} - 2\tau = \frac{\epsilon}{2sn} - \frac{\epsilon^2}{20sn^2 + 2\epsilon} \geq \frac{\epsilon}{4sn} \tag{2}$$

Eventually we have

$$\Pr[\theta_{V,a}(x) = a] \geq \Pr[V \sqsubseteq x[1, n-1]] \cdot \Pr[x_{I_{V \sqsubseteq x}+1} = a \mid V \sqsubseteq x[1, n-1]]$$

$$\geq \left(1 - \frac{\epsilon}{20sn^2 + 2\epsilon}\right) \frac{\epsilon^2}{4sn} \tag{3}$$

is polynomially large. Query $\chi_{V,a}$ is legitimate and feasible. ∎

The correctness of the algorithm is based on the intuition that the query result $\mathbf{E}\chi_{V,a_+}$ of $a_+ \in U_{\ell+1}$ should be greater than that of $a_- \notin U_{\ell+1}$ and the difference is large enough to tolerate the noise from the oracle. To prove this, we first consider the exact learning case. Define an infinite string $U' = U[1, \ell]U[\ell + 2, L]U^\infty_{\ell+1}$ and let $x' = x\Sigma^\infty$ be the extension of $x$ obtained by padding it on the right with an infinite string generated from the same distribution as $x$. Let $Q(j, i)$ be the probability that the largest $g$ such that $U'[1, g] \sqsubseteq x'[1, i]$ is $j$, or formally, $Q(j, i) = \Pr[U'[1, j] \sqsubseteq x'[1, i] \wedge U'[1, j+1] \not\sqsubseteq x'[1, i]]$.

**Lemma 2 (in the main paper)** *Under element-wise independent and identical distributions over instance space $\mathcal{I} = \Sigma^n$, concept class $\text{Ш}$ is exactly identifiable with $O(sn)$ conditional statistical queries from STAT$(\text{Ш}, \mathcal{D})$ at tolerance*

$$\tau' = \frac{1}{5}Q(L - 1, n - 1)$$

**Proof** If the algorithm doesn't halt, $U$ has not been completely recovered and $\ell < L$. By assumption, $V = U[1, \ell]$. If $V \not\sqsubseteq x[1, n-1]$ then $x$ must be a negative example and $\chi_{V,a}(x, y) = 0$. Hence $\chi_{V,a}(x, y) = 1$ if and only if $V \sqsubseteq x[1, n-1]$ and $y = +1$.

Let random variable $J$ be the largest value for which $U'[1, J]$ is a subsequence of $x[1, n-1]$. Consequently, $\Pr[J = j] = Q(j, n-1)$.

If $a \in U_{\ell+1}$, then $y = +1$ if and only if $J \geq L - 1$. Thus we have

$$\mathbf{E}\chi_{V,a} = \sum_{j=L-1}^{n-1} Q(j, n-1)$$

If $a \notin U_{\ell+1}$, then $y = +1$ if and only if $U \sqsubseteq x[1, I_{V \sqsubseteq x}]x[I_{V \sqsubseteq x} + 2, n]$. Since elements in a string are i.i.d., $\Pr[U \sqsubseteq x[1, I_{V \sqsubseteq x}]x[I_{V \sqsubseteq x} + 2, n]] = \Pr[U'[1, L] \sqsubseteq x[1, n-1]]$, which is exactly $\Pr[J \geq L]$. Thus we have

$$\mathbf{E}\chi_{V,a} = \sum_{j=L}^{n-1} Q(j, n-1)$$

The difference between these two values is $Q(L-1, n-1)$. In order to distinguish the target $U_{\ell+1}$ from other symbols, the query tolerance can be set to one fifth of the difference. The alphabet $\Sigma$ will be separated into two clusters by the results of $\mathbf{E}_{\chi_{V,a}}$: $U_{\ell+1}$ and the other symbols. The maximum difference (variance) inside a cluster is smaller than the minimum difference (gap) between the two clusters, making them distinguishable. As a consequence $s$ statistical queries for each prefix of $U$ suffice to learn $U$ exactly. ∎

Lemma 2 indicates bounding the quantity $Q(L-1, n-1)$ is the key to the tolerance for PAC learning. Unfortunately, the distribution $\{Q(j, i)\}$ doesn't seem of any strong properties we know of providing a polynomial lower bound. Instead we introduce new quantity $R(j, i) = \Pr[U'[1, j] \sqsubseteq x'[1, i] \wedge U'[1, j] \not\sqsubseteq x'[1, i-1]]$ being the probability that the smallest $g$ such that $U'[1, j] \sqsubseteq x'[1, g]$ is $i$. Now we show the strong unimodality of distribution $\{R(j, i)\}$. Denote $p_j = \Pr[x_i \in U'_j]$.

**Lemma 6** *The convolution of two strongly unimodal discrete distributions is strongly unimodal.*

**Proof** The proof is obvious from the definition of strong unimodality and the associativity of convolution. Let $H_3 = H_2 * H_1$ be the convolution of two strongly unimodal distributions $H_1$ and $H_2$. For any unimodal distribution $P_1$, let $P_2 = H_1 * P_1$ be the convolution of $H_1$ and $P_1$. Because of the strong unimodality of distribution $H_1$, $P_2$ is a unimodal distribution. Also because of the strong unimodality of distribution $H_2$, the convolution of $H_3$ and $P_1$, $H_3 * P_1 = H_2 * H_1 * P_1 = H_2 * P_2$ is a unimodal distribution. Since $P_1$ can be an arbitrary unimodal distribution, $H_3$ is strongly unimodal according to the definition of strong unimodality. ∎

Previous work [10] provided a useful equivalent statement on strong unimodality of a distribution.

**Lemma 7 ([10])** *Distribution $\{H(i)\}$ is strongly unimodal if and only if $H(i)$ is log-concave. That is,*

$$H(i)^2 \geq H(i+1) \cdot H(i-1)$$

*for all $i$.*

Since a distribution with all mass at zero is unimodal, an immediate consequence is

**Corollary 3** *A strongly unimodal distribution is unimodal.*

We now prove the strong unimodality of distribution $\{R(j, i)\}$.

**Lemma 8** *For any fixed $j$, distribution $\{R(j, i)\}$ is strongly unimodal with respect to $i$.*

**Proof** This proof can be done by induction on $j$ as follows.

*Basis*: For $j = 1$, it is obvious that $\{R(1, i)\} = \{(1-p_1)^{i-1}p_1\}$ is a geometric distribution, which is strongly unimodal. According to Lemma 7, this is due to $R^2(1, i) = R(1, i-1) \cdot R(1, i+1)$ for all $i > 1$.

*Inductive step*: For $j > 1$, assume by induction $\{R(j-1, i)\}$ is strongly unimodal. Based on the definition of $R(j, i)$, we have

$$R(j, i) = \sum_{k=j-1}^{i-1} \left( R(j-1, k) \cdot (1-p_j)^{i-k-1}p_j \right) \tag{4}$$

Thus $R(j, i)$ is the convolution of distribution $\{R(j-1, i)\}$ and distribution $\{(1-p_j)^{i-1}p_j\}$, a geometric distribution just proved to be strongly unimodal. By assumption, $\{R(j-1, i)\}$ is strongly unimodal. From Lemma 6, distribution $\{R(j, i)\}$ is also strongly unimodal.

*Conclusion*: For any fixed $j$, distribution $\{R(j, i)\}$ is strongly unimodal with respect to $i$. ∎

Combining Lemma 8 with Corollary 3, we have

**Corollary 4** *For any fixed $j$, distribution $\{R(j, i)\}$ is unimodal with respect to $i$.*

**Lemma 9** *Denote by $N(j)$ the mode of $\{R(j, i)\}$, then $N(j)$ is strictly increasing with respect to $j$. That is, for any $j > 1$, $N(j) > N(j-1)$.*

**Proof** According to Equation 4, $R(j, i)$ is the convolution of distribution $\{R(j-1, i)\}$ and distribution $\{(1 - p_j)^{i-1} p_j\}$ so

$$R(j, i) = \sum_{k=j-1}^{i-1} \left( R(j-1, k) \cdot (1 - p_j)^{i-k-1} p_j \right)$$

and

$$R(j, i+1) = \sum_{k=j-1}^{i} \left( R(j-1, k) \cdot (1 - p_j)^{i-k} p_j \right)$$

Hence, we get

$$R(j, i+1) = p_j R(j-1, i) + (1 - p_j) R(j, i) \tag{5}$$

Denote by $\Delta R(j, i)$ the difference $R(j, i) - R(j, i-1)$. From Equation 5, we have

$$\Delta R(j, i+1) = p_j \Delta R(j-1, i) + (1 - p_j) \Delta R(j, i) \tag{6}$$

For any $j \geq 1$, we have $R(j, 1) \geq R(j, 0) = 0$ or $\Delta R(j, 1) \geq 0$. From the definition of $N(j)$, $N(j)$ must be at least $j$ and for any $i \leq N(j-1)$, the difference $\Delta R(j-1, i)$ is non-negative. Hence, if $\Delta R(j, i)$ is non-negative, then $\Delta R(j, i+1)$ is non-negative for Equation 6. So inductively, for any $i \leq N(j-1) + 1$, we always have $\Delta R(j, i) \geq 0$. Recall that we define the mode of a distribution with multiple modes as the one with the largest index, thus $N(j) > N(j-1)$. ∎

With the strong unimodality of distribution $\{R(j, i)\}$, we are able to present the PAC learnability of concept class Ш in the statistical query model.

**Theorem 1 (in the main paper)** *Under element-wise independent and identical distributions over instance space $\mathcal{I} = \Sigma^n$, concept class Ш is approximately identifiable with $O(sn)$ conditional statistical queries from STAT$(Ш, \mathcal{D})$ at tolerance*

$$\tau = \frac{\epsilon^2}{40sn^2 + 4\epsilon}$$

*or with $O(sn)$ statistical queries from STAT$(Ш, \mathcal{D})$ at tolerance*

$$\bar{\tau} = \left( 1 - \frac{\epsilon}{20sn^2 + 2\epsilon} \right) \frac{\epsilon^4}{16sn(10sn^2 + \epsilon)}$$

**Proof** From Lemma 5, statistical query $\chi_{V,a}$ is legitimate and feasible at tolerance $\tau = \epsilon^2 / (40sn^2 + 4\epsilon)$ and our error parameter must be set to $\epsilon/2$ in order to have Inequality 2.

We modify the statistical query algorithm to make an initial statistical query with tolerance $\tau = \epsilon^2 / (40sn^2 + 4\epsilon)$ to estimate $\Pr[y = +1]$. If the answer is $\leq \epsilon/2 - \tau$, then $\Pr[y = +1] \leq \epsilon/2$ and the algorithm outputs a hypothesis that all examples are negative. If the answer is $\geq 1 - \epsilon/2 + \tau$, then $\Pr[y = +1] \geq 1 - \epsilon/2$ and the algorithm outputs a hypothesis that all examples are positive.

Otherwise, $\Pr[y = +1]$ and $\Pr[y = -1]$ are both at least $\epsilon/2 - 2\tau$. We then do another statistical query at tolerance $\tau$ to estimate $\Pr[y = +1 \mid V \sqsubseteq x]$. Since $V \sqsubseteq x$ is a necessary condition of positivity, $\Pr[V \sqsubseteq x]$ must be at least $\Pr[y = +1] \geq \epsilon/2 - 2\tau$ and this statistical query is legitimate and feasible. If the answer is $\geq 1 - \epsilon/2 + \tau$, then $\Pr[y = +1 \mid V \sqsubseteq x] \geq 1 - \epsilon/2$. The algorithm outputs a hypothesis that all strings $x$ such that $V \sqsubseteq x$ are positive and all strings $x$ such that $V \not\sqsubseteq x$ are negative because $\Pr[y = -1 \mid V \not\sqsubseteq x] = 1$. If $\ell = L$, $\Pr[y = +1 \mid V \sqsubseteq x]$ must be 1 and the algorithm halts. Otherwise, $\ell < L$ and the first statistical query algorithm is used. We now show that $Q(L-1, n-1) \geq 5\tau$, establishing the bound on the query tolerance.

Let random variable $I$ be the smallest value for which $U'[1, L]$ is a subsequence of $x'[1, I]$. Based on the definition of $R(j, i)$, we have $\Pr[I = i] = R(L, i)$. String $x$ is a positive example if and only if $U'[1, L] \sqsubseteq x'[1, n]$, which is exactly $I \leq n$. As a consequence,

$$\Pr[y = +1] = \sum_{i=L}^{n} R(L, i) \tag{7}$$

From Corollary 4, distribution $\{R(L, i)\}$ is unimodal and assume its mode is $N(L)$. If $n \leq N(L)$ then $R(L, n)$ is at least as large as every term in the sum $\Pr[y = +1] = \sum_{i=L}^{n} R(L, i)$. Hence we get

$$R(L, n) \geq \frac{\epsilon - 4\tau}{2(n - L + 1)} \geq \frac{\epsilon - 4\tau}{2n} \geq \frac{5\epsilon^2}{40sn^2 + 4\epsilon} = 5\tau$$

If $n > N(L)$, according to Lemma 9, for any $j \leq L$ we have $n > N(j)$. That is, for any $j \leq L$, we have $R(j, n) \geq R(j, n + 1)$.

From Equation 5,

$$R(j, n + 1) = p_j R(j - 1, n) + (1 - p_j) R(j, n)$$

so

$$p_j R(j - 1, n) + (1 - p_j) R(j, n) \leq R(j, n)$$
$$= p_j R(j, n) + (1 - p_j) R(j, n)$$

We then have

$$R(j - 1, n) \leq R(j, n)$$

This holds for any $j \leq L$ so $R(j, n)$ is non-decreasing with respect to $j$ when $n > N(L)$. Inductively we get $R(L, n) \geq R(j, n)$ for any $j \leq L$.

Because $U'[1, L] \not\sqsubseteq x[1, n - 1]$ is a necessary condition of $y = -1$ and

$$\Pr[U'[1, L] \not\sqsubseteq x[1, n - 1]] = \sum_{j=0}^{L-1} Q(j, n - 1)$$

we get

$$\sum_{j=0}^{L-1} Q(j, n - 1) \geq \Pr[y = -1] \geq \frac{\epsilon - 4\tau}{2}$$

Note that $R(j, n) = p_j Q(j - 1, n - 1)$, then

$$\sum_{j=1}^{L} \frac{R(j, n)}{p_j} \geq \frac{\epsilon - 4\tau}{2}$$

Since

$$\Pr[y = +1] \geq \frac{\epsilon - 4\tau}{2} > 0$$

from Inequality 2, we must have $p_j \geq \epsilon/(4sn)$ for all $j$. Then we have

$$\frac{4sn}{\epsilon} \sum_{j=1}^{L} R(j, n) \geq \sum_{j=1}^{L} \frac{R(j, n)}{p_j} \geq \frac{\epsilon - 4\tau}{2}$$

Because $R(L, n) \geq R(j, n)$ for any $j \leq L$, we get

$$\frac{4sn}{\epsilon} L R(L, n) \geq \frac{\epsilon - 4\tau}{2}$$

and

$$R(L, n) \geq \frac{(\epsilon - 4\tau)\epsilon}{8sn^2} = \frac{5\epsilon^2}{40sn^2 + 4\epsilon} = 5\tau$$

Finally, we have

$$Q(L, n) = (1 - p_{L+1}) Q(L, n - 1) + p_L Q(L - 1, n - 1)$$

$$\geq p_L Q(L - 1, n - 1) = R(L, n) \geq \frac{5\epsilon^2}{40sn^2 + 4\epsilon}$$

That is, $Q(L - 1, n - 1) \geq 5\tau$. For Lemma 2, we have $\tau = \epsilon^2/(40sn^2 + 4\epsilon)$. Inferring $\bar{\tau}$ from $\tau$ is trivial. Define general statistical query

$$\bar{\chi}_{V,a}(x, y) = \begin{cases} (y + 1)/2 & \text{if } \theta_{V,a}(x) = a \\ 0 & \text{if } \theta_{V,a}(x) \neq a \end{cases} \tag{8}$$

Then for any $a$, the expected query result

$$\mathbf{E}\bar{\chi}_{V,a} = \Pr[\theta_{V,a}(x) = a] \cdot \mathbf{E}\chi_{V,a} + 0$$

and the difference between $\mathbf{E}\bar{\chi}_{V,a} \mid a \in U_{\ell+1}$ and $\mathbf{E}\bar{\chi}_{V,a} \mid a \notin U_{\ell+1}$ is $5\tau \cdot \Pr[\theta_{V,a}(x) = a]$. Hence, from Inequality 3,

$$\bar{\tau} = \left(1 - \frac{\epsilon}{20sn^2 + 2\epsilon}\right)\frac{\epsilon^4}{16sn(10sn^2 + \epsilon)}$$

This completes the proof. ∎

# B   Proof of Theorem 2

To calculate the tolerance for PAC learning, we first consider the exact learning tolerance. Let $x'$ be an infinite string generated by the Markov chain defined in Section 4. For any $0 \le \ell \le L$, we define quantity $R_\ell(j, i)$ by

$$R_\ell(j, i) = \Pr[u[\ell+1, \ell+j] \sqsubseteq x'[m+1, m+i] \wedge u[\ell+1, \ell+j] \not\sqsubseteq x'[m+1, m+i-1] \mid x'_m = u_\ell]$$

Intuitively, $R_\ell(j, i)$ is the probability that the smallest $g$ such that $u[\ell+1, \ell+j] \sqsubseteq x'[m+1, m+g]$ is $i$, given $x'_m = u_\ell$. We have the following conclusion on the exact learning tolerance.

**Lemma 3 (in the main paper)** *Under Markovian string distributions over instance space $\mathcal{I} = \Sigma^{\le n}$, given $\Pr[|x| = k] \ge t > 0$ for $\forall 1 \le k \le n$ and $\min\{M(\cdot, \cdot), \pi_0(\cdot)\} \ge c > 0$, the concept class $\sqcup\!\sqcup$ is exactly identifiable with $O(sn^2)$ conditional statistical queries from STAT($\sqcup\!\sqcup, \mathcal{D}$) at tolerance*

$$\tau' = \min_{0 \le \ell < L}\left\{\frac{1}{3(n-h)}\sum_{k=h+1}^{n} R_{\ell+1}(L - \ell - 1, k - h - 1)\right\}$$

**Proof**  If the algorithm doesn't halt, $u$ has not been completely recovered and $\ell < L$. Again, we calculate the difference of $\Psi_{v,a}$ between the cases $a_+ = u_{\ell+1}$ and $a_- \ne u_{\ell+1}$.

For $a_- \ne u_{\ell+1}$, let $p_j$ denote the probability that the first passage time from $a_-$ to $u_{\ell+1}$ is equal to $j$. Notice that

$$\mathbf{E}\chi_{v,a_-,k} = \sum_{j=1}^{k-h-1}\left(p_j\sum_{i=0}^{k-h-1-j} R_{\ell+1}(L-\ell-1, i)\right)$$

$$\le \sum_{j=1}^{k-h-1}\left(p_j\sum_{i=0}^{k-h-2} R_{\ell+1}(L-\ell-1, i)\right)$$

We get

$$\mathbf{E}\bar{\chi}_{v,a_-,k} \le \sum_{i=0}^{k-h-2} R_{\ell+1}(L-\ell-1, i)$$

For $a_+ = u_{\ell+1}$, we have

$$\mathbf{E}\chi_{v,a_+,k} = \sum_{i=0}^{k-h-1} R_{\ell+1}(L-\ell-1, i)$$

Summing up all the items, we can get the difference

$$\Psi_{v,a_+} - \Psi_{v,a_-} = \sum_{k=h+1}^{n}\left(\mathbf{E}\chi_{v,a_+,k} - \mathbf{E}\chi_{v,a_-,k}\right)$$

$$\ge \sum_{k=h+1}^{n}\left(\sum_{i=0}^{k-h-1} R_{\ell+1}(L-\ell-1, i) - \sum_{i=0}^{k-h-2} R_{\ell+1}(L-\ell-1, i)\right)$$

$$= \sum_{k=h+1}^{n} R_{\ell+1}(L-\ell-1, k-h-1)$$

In order to distinguish the target $u_{\ell+1}$ from other symbols, the query tolerance can be set to one third of the difference so that the symbol with largest query result must be $u_{\ell+1}$. Thus the overall tolerance for $\Psi_{v,a}$ is $\sum_{k=h+1}^{n} R_{\ell+1}(L-\ell-1, k-h-1)/3$. Since $\Psi_{v,a}$ is the expectation sum of $(n-h)$ statistical queries, we can evenly distribute the overall tolerance on each $\chi_{v,a,k}$. So the final tolerance on each statistical query is

$$\tau' = \min_{0 \leq \ell < L} \left\{ \frac{1}{3(n-h)} \sum_{k=h+1}^{n} R_{\ell+1}(L-\ell-1, k-h-1) \right\}$$

Taking minimum over $0 \leq \ell < L$ is because $h$ depends on $\ell$ and the tolerance needs to be independent of $h$. As a consequence $sn$ statistical queries for each prefix of $U$ suffice to learn $U$ exactly. ∎

We then show how to choose a proper $h$ from $[0, n-1]$.

**Lemma 10** *Under Markovian string distributions over instance space $\mathcal{I} = \Sigma^{\leq n}$, given $\Pr[|x| = k] \geq t > 0$ for $\forall 1 \leq k \leq n$ and $\min\{M(\cdot, \cdot), \pi_0(\cdot)\} \geq c > 0$, the conditional statistical query $\chi_{v,a,k}$ is legitimate and feasible at tolerance*

$$\tau = \frac{\epsilon}{3n^2 + 2n + 2}$$

**Proof** First of all, the function $\chi_{v,a,k}$ computes a binary mapping from labeled examples $(x, y)$ to $\{0, 1\}$ and satisfies the definition of a statistical query. Under the given conditions, $\chi_{v,a,k}$ returns $0$ if $x$ is a negative example ($y = -1$) or returns $1$ if $x$ is a positive example ($y = +1$).

From Lemma 1, evaluating the relation $v \sqsubseteq x$ and meanwhile determining $I_{v \sqsubseteq x}$ is feasible in time $O(n)$. Since $|x| \leq n$, determining $|x|$ also takes $O(n)$ time. Thus, $\chi_{v,a,k}(x, y)$ and then $\Psi_{v,a}(x, y)$ can be efficiently evaluated.

According to the Markov property and the independence between string length and symbols in a string, we have

$$\Pr[I_{v \sqsubseteq x} = h, \ x_{h+1} = a \text{ and } |x| = k]$$
$$= \Pr[I_{v \sqsubseteq x} = h] \cdot \Pr[x_{h+1} = a \mid I_{v \sqsubseteq x} = h] \cdot \Pr[|x| = k]$$
$$\geq \Pr[I_{v \sqsubseteq x} = h] \cdot c \cdot t$$

The only problem left is to make sure $\Pr[I_{v \sqsubseteq x} = h]$ is polynomially large. Obviously this can't be guaranteed for all $h$ between $\ell$ and $n-1$ so $h$ must be chosen carefully. We now show there must be such an $h$.

We make an initial statistical query with tolerance $\epsilon/(3n^2 + 2n + 2)$ to estimate $\Pr[y = +1]$. If the answer is $\leq (3n^2 + 2n + 1)\epsilon/(3n^2 + 2n + 2)$, then $\Pr[y = +1] \leq \epsilon$ and the algorithm outputs a hypothesis that all examples are negative. Otherwise, $\Pr[y = +1]$ is at least $(3n^2 + 2n)\epsilon/(3n^2 + 2n + 2)$, and the statistical queries $\{\chi_{v,a,k}\}$ are used. Since

$$\Pr[y = +1] = \sum_{h=\ell}^{n-1} \Pr[y = +1 \wedge I_{v \sqsubseteq x} = h] \tag{9}$$

There must be at least one $h$ so that

$$\Pr[y = +1 \wedge I_{v \sqsubseteq x} = h] \geq \frac{1}{n-h} \Pr[y = +1]$$
$$\geq \frac{1}{n} \Pr[y = +1]$$
$$\geq \frac{1}{n} \cdot \frac{(3n^2 + 2n)\epsilon}{3n^2 + 2n + 2}$$
$$= \frac{(3n + 2)\epsilon}{3n^2 + 2n + 2}$$

As
$$\Pr[y = +1 \wedge I_{v \sqsubseteq x} = h] = \Pr[y = +1 \mid I_{v \sqsubseteq x} = h] \cdot \Pr[I_{v \sqsubseteq x} = h]$$
both $\Pr[y = +1 \mid I_{v \sqsubseteq x} = h]$ and $\Pr[I_{v \sqsubseteq x} = h]$ must be at least $(3n + 2)\epsilon/(3n^2 + 2n + 2)$. This means there must be some $h$ making our statistical queries legitimate.

We now show how to determine a proper value of $h$. We can do a statistical query

$$\chi'_h(x, y) = \frac{1}{2}(y + 1) \cdot \mathbb{1}_{\{I_{v \sqsubseteq x} = h\}} \tag{10}$$

for each $h$ from $\ell$ to $n - 1$, where $\mathbb{1}_{\{\pi\}}$ represents the 0-1 truth value of the predicate $\pi$. It is easy to see $\mathbf{E}\chi'_h = \Pr[y = +1 \wedge I_{v \sqsubseteq x} = h]$. According to our analysis above and due to the noise of the statistical query, there must be at least one $h$ such that the answer is $\geq (3n + 1)\epsilon/(3n^2 + 2n + 2)$. If we choose such an $h$, it is guaranteed to have

$$\Pr[y = +1 \wedge I_{v \sqsubseteq x} = h] \geq \frac{3n\epsilon}{3n^2 + 2n + 2}$$

so that

$$\Pr[I_{v \sqsubseteq x} = h] \geq \frac{3n\epsilon}{3n^2 + 2n + 2}$$

and

$$\Pr[y = +1 \mid I_{v \sqsubseteq x} = h] \geq \frac{3n\epsilon}{3n^2 + 2n + 2} \tag{11}$$

After at most $n$ statistical queries $\{\chi'_h\}$, we can determine the value of $h$ in query $\chi_{v,a,k}$. Thus statistical queries $\{\chi_{v,a,k}\}$ and $\Psi_{v,a}$ are legitimate and feasible. $\blacksquare$

Below is the proof of Theorem 2.

**Theorem 2 (in the main paper)** *Under Markovian string distributions over instance space $\mathcal{I} = \Sigma^{\leq n}$, given $\Pr[|x| = k] \geq t > 0$ for $\forall 1 \leq k \leq n$ and $\min\{M(\cdot, \cdot), \pi_0(\cdot)\} \geq c > 0$, concept class $\sqcup$ is approximately identifiable with $O(sn^2)$ conditional statistical queries from STAT($\sqcup, \mathcal{D}$) at tolerance*
$$\tau = \frac{\epsilon}{3n^2 + 2n + 2}$$
*or with $O(sn^2)$ statistical queries from STAT($\sqcup, \mathcal{D}$) at tolerance*
$$\bar{\tau} = \frac{3ctn\epsilon^2}{(3n^2 + 2n + 2)^2}$$

**Proof** From Lemma 10, statistical queries $\{\chi_{v,a,k}\}$ and $\Psi_{v,a}$ are legitimate and feasible at tolerance $\epsilon/(3n^2 + 2n + 2)$.

We modify the statistical query algorithm to make an initial statistical query with tolerance $\epsilon/(3n^2 + 2n + 2)$ to estimate $\Pr[y = +1]$. If the answer is $\leq (3n^2 + 2n + 1)\epsilon/(3n^2 + 2n + 2)$, then $\Pr[y = +1] \leq \epsilon$ and the algorithm outputs a hypothesis that all examples are negative. Otherwise, $\Pr[y = +1]$ is at least $(3n^2 + 2n)\epsilon/(3n^2 + 2n + 2)$.

We then do another statistical query with tolerance $\epsilon/(3n^2 + 2n + 2)$ to estimate $\Pr[y = +1 \mid v \sqsubseteq x]$. Since $v \sqsubseteq x$ is a necessary condition of positivity, $\Pr[v \sqsubseteq x]$ must be at least $\Pr[y = +1] \geq (3n^2 + 2n)\epsilon/(3n^2 + 2n + 2)$ and this statistical query is legitimate and feasible. If the answer is $\geq 1 - (3n^2 + 2n)\epsilon/(3n^2 + 2n + 2)$, then $\Pr[y = +1 \mid v \sqsubseteq x] \geq 1 - \epsilon$. The algorithm outputs a hypothesis that all strings $x$ such that $v \sqsubseteq x$ are positive and all strings $x$ such that $v \not\sqsubseteq x$ are negative because $\Pr[y = -1 \mid v \not\sqsubseteq x] = 1$. If $\ell = L$, $\Pr[y = +1 \mid v \sqsubseteq x]$ must be 1 and the algorithm halts. Otherwise, $\ell < L$ and the first statistical query algorithm is used.

From the proof for Lemma 10, we then use $O(n)$ statistical queries

$$\chi'_h(x, y) = \frac{1}{2}(y + 1) \cdot \mathbb{1}_{\{I_{v \sqsubseteq x} = h\}}$$

to find an $h$ such that Inequality 11 holds:

$$\Pr[y = +1 \mid I_{v \sqsubseteq x} = h] \geq \frac{3n\epsilon}{3n^2 + 2n + 2}$$

Similarly, let $q_j$ denote the probability that the first passage time from $u_\ell$ to $u_{\ell+1}$ is equal to $j$. Notice that

$$\Pr[y = +1 \mid I_{v \sqsubseteq x} = h] \leq \sum_{j=1}^{n-h} \left( q_j \sum_{i=0}^{n-h-j} R_{\ell+1}(L - \ell - 1, i) \right)$$

We have

$$\frac{3n\epsilon}{3n^2 + 2n + 2} \leq \Pr[y = +1 \mid I_{v \sqsubseteq x} = h]$$

$$\leq \sum_{j=1}^{n-h} \left( q_j \sum_{i=0}^{n-h-j} R_{\ell+1}(L - \ell - 1, i) \right)$$

$$\leq \sum_{j=1}^{n-h} \left( q_j \sum_{i=0}^{n-h-1} R_{\ell+1}(L - \ell - 1, i) \right)$$

$$\leq \sum_{i=0}^{n-h-1} R_{\ell+1}(L - \ell - 1, i)$$

$$= \sum_{k=h+1}^{n} R_{\ell+1}(L - \ell - 1, k - h - 1)$$

From Lemma 3, the conditional tolerance is

$$\tau = \min_{0 \leq \ell < L} \left\{ \frac{1}{3(n-h)} \sum_{k=h+1}^{n} R_{\ell+1}(L - \ell - 1, k - h - 1) \right\} \geq \frac{\epsilon}{3n^2 + 2n + 2}$$

Similar to the proof of Theorem 1, define general statistical query

$$\bar{\chi}_{v,a,k}(x, y) = \begin{cases} (y+1)/2 & \text{if } I_{v \sqsubseteq x} = h, \ x_{h+1} = a \text{ and } |x| = k \\ 0 & \text{otherwise} \end{cases} \tag{12}$$

and

$$\bar{\Psi}_{v,a} = \sum_{k=h+1}^{n} \mathbf{E}\bar{\chi}_{v,a,k}(x, y) \tag{13}$$

Then the general tolerance $\bar{\tau}$ can be easily inferred from the conditional tolerance $\tau$:

$$\bar{\tau} = \frac{3ctn\epsilon^2}{(3n^2 + 2n + 2)^2}$$

Considering we have used $n$ statistical queries to determine $h$, $(s + 1)n$ statistical queries for each prefix of $u$ suffice to PAC learn $u$. This completes the proof. ∎

## C  A constrained generalization to learning shuffle ideals under product distributions

In this section we generalize the idea in Section 3 to learning the extended class of shuffle ideals when example strings are drawn from a product distribution. For any augmented string $V \in (\Sigma^\cup)^{\leq n}$, any symbol $a \in \Sigma$ and any integer $h \in [0, n-1]$, define

$$\tilde{\chi}_{V,a,h}(x, y) = \frac{1}{2}(y+1) \quad \text{given } I_{V \sqsubseteq x} = h \text{ and } x_{h+1} = a$$

where $y = c(x)$ is the label of $x$. Again the algorithm uses query $\Pr[y = +1 \mid V \sqsubseteq x]$ to tell whether it is time to halt. As before, let $V$ be the partial pattern we have learned and the algorithm starts with $V = \lambda$. For $1 \leq i \leq n$ and $1 \leq j \leq L$, define probability $\tilde{Q}(j, i)$ as below.

$$\tilde{Q}(j, i) = \begin{cases} \Pr[U[L-j+1, L] \sqsubseteq x[n-i+1, n] \wedge U[L-j, L] \not\sqsubseteq x[n-i+1, n]] & \text{if } 1 \leq j < L \\ \Pr[U \sqsubseteq x[n-i+1, n]] & \text{if } j = L \end{cases}$$

**Lemma 11** *Under product distributions over instance space $\mathcal{I} = \Sigma^n$, given $\Pr[x_i = a] \geq t > 0$ for $\forall 1 \leq i \leq n$ and $\forall a \in \Sigma$, concept class III is exactly identifiable with $O(sn)$ conditional statistical queries from STAT(III, $\mathcal{D}$) at tolerance*

$$\tau' = \frac{1}{5} \min \left\{ \tilde{Q}(L-1, n-1), \min_{1 \leq \ell \leq L} \max_{\ell \leq h \leq n-1} \tilde{Q}(L-\ell-1, n-h-1) \right\}$$

**Proof** If the algorithm doesn't halt, $U$ has not been completely recovered and $\ell < L$. As before, we calculate the difference of $\mathbf{E}\tilde{\chi}_{V,a,h}$ between the cases $a_+ \in U_{\ell+1}$ and $a_- \notin U_{\ell+1}$.

When $\ell = 0$ and $V = \lambda$, the value of $I_{V \sqsubseteq x}$ must be 0 so $h$ is fixed to be 0 in the query. For symbol $a_+ \in U_1$, we have

$$\mathbf{E}\tilde{\chi}_{\lambda, a_+, 0} = \tilde{Q}(L-1, n-1) + \tilde{Q}(L, n-1)$$

and for symbol $a_- \notin U_1$,

$$\mathbf{E}\tilde{\chi}_{\lambda, a_-, 0} = \tilde{Q}(L, n-1)$$

Taking one fifth of the difference gives the tolerance $\tilde{Q}(L-1, n-1)/5$ for $\ell = 0$.

When $1 \leq \ell < L$ and $V = U[1, \ell]$, we have for symbol $a_+ \in U_{\ell+1}$,

$$\mathbf{E}\tilde{\chi}_{V, a_+, h} = \sum_{j=L-\ell-1}^{L} \tilde{Q}(j, n-h-1)$$

and for symbol $a_- \notin U_{\ell+1}$,

$$\mathbf{E}\tilde{\chi}_{V, a_-, h} = \sum_{j=L-\ell}^{L} \tilde{Q}(j, n-h-1)$$

Again taking one fifth of the difference gives the tolerance $\tilde{Q}(L-\ell-1, n-h-1)/5$. For a fixed $1 \leq \ell < L$, tolerance $\max_{\ell \leq h \leq n-1} \tilde{Q}(L-\ell-1, n-h-1)/5$ is enough to learn $U_{\ell+1}$ exactly. Taking the minimum tolerance among all $0 \leq \ell < L$ gives the overall tolerance in the statement. As a consequence $s$ statistical queries for each prefix of $U$ suffice to learn $U$ exactly. ∎

A more complicated algorithm is needed to PAC learn shuffle ideals under product distributions. We first define two additional simple queries:

$$\chi'_{V,a,h,i}(x,y) = \mathbb{1}_{\{x_{h+i}=a\}} \quad \text{given } I_{V \sqsubseteq x} = h$$
$$\chi^+_{V,a,h,i}(x,y) = \mathbb{1}_{\{x_{h+i}=a\}} \quad \text{given } I_{V \sqsubseteq x} = h \text{ and } y = +1$$

whose expectations serve as empirical estimators for the distributions of the symbol at the next $i$-th position over all strings ($\chi'_{V,a,i}$) and over all positive strings ($\chi^+_{V,a,i}$), both conditioned on $I_{V \sqsubseteq x} = h$. Below is how the algorithm works, with $\bar{\epsilon}_{g+1}$ and $\epsilon'$ to be decided later in the proof.

First an initial query to estimate probability $\Pr[y = +1 \mid V \sqsubseteq x]$ is made. The algorithm will classify all strings such that $V \sqsubseteq x$ negative if the answer is close to 0, or positive if the answer is close to 1. To ensure the legitimacy and feasibility of the algorithm, we make another initial query to estimate the probability $\Pr[I_{V \sqsubseteq x} = h]$ for each $h$. The algorithm then excludes the low-probability cases such that any of the excluded ones happens with probability lower than $\epsilon/2$. Thus we only need to consider the cases with polynomially large $\Pr[I_{V \sqsubseteq x} = h]$ and learn the target ideal within error $\epsilon/2$. Otherwise, let $P(+|a,h)$ denote $\mathbf{E}\tilde{\chi}_{V,a,h}$ and we make a statistical query to estimate $P(+|a,h)$ for each $a \in \Sigma$. If the difference $P(+|a_+,h) - P(+|a_-,h)$, where $a_+$ is in the next element of $U$ and $a_-$ is not, is large enough for some $h$, then the results of queries for $P(+|a,h)$ will form two distinguishable clusters, where the maximum difference inside one cluster is smaller than the minimum gap between them, so that we are able to learn the next element in $U$.

Otherwise, for all $h$ with nonnegligible $\Pr[I_{V \sqsubseteq x} = h]$, the difference $P(+|a_+,h) - P(+|a_-,h)$ is very small and we will show that there is an interval starting from index $h+1$ which we can skip with little risk for each case when $I_{V \sqsubseteq x} = h$. Problematic cases leading to misclassification will happen with very small probability within this interval. We are safe to skip the whole interval and move on.

1. Estimate probability $\Pr[y = +1 \mid V \sqsubseteq x]$ at tolerance $\epsilon'/3$. If the answer is $\leq 2\epsilon'/3$, classify all strings $x$ such that $V \sqsubseteq x$ as negative and backtrack on the classification tree. If the answer is $\geq 1 - 2\epsilon'/3$, classify all strings $x$ such that $V \sqsubseteq x$ as positive and backtrack. If the number of intervals skipped on the current path exceeds $C$, classify all strings $x$ such that $V \sqsubseteq x$ as positive and backtrack. Otherwise go to Step 2

2. For each $h$ with nonnegligible $\Pr[I_{V \sqsubseteq x} = h]$, estimate $\mathbf{E}\chi_{V,a,h}$ at tolerance $\tau_1 = \bar{\epsilon}_{g+1}^2/384$ for each $a \in \Sigma$. Go to Step 3.

3. If the results for some $h$ produce two distinguishable clusters, where the maximum difference inside one cluster is $\leq 4\tau_1$ while the minimum gap between two clusters is $> 4\tau_1$, then the set of all the symbols that belong to the cluster with larger query results is the next element in $U$. Update $V$ and go to Step 1. Otherwise, branch the classification tree. For each $h$, let $k \leftarrow 1$ and $T \leftarrow 1$. Go to Step 4.

4. For each $a \in \Sigma$, estimate $\mathbf{E}\chi'_{V,a,h,k}$ and $\mathbf{E}\chi^+_{V,a,h,k}$ at tolerance $\tau_2 = \bar{\epsilon}_{g+1}/(8sn)$ so that we will have estimators $\widehat{\mathcal{D}}_k(h)$ and $\widehat{\mathcal{D}}_k^+(h)$. Go to Step 5.

5. $T \leftarrow (1 - \|\widehat{\mathcal{D}}_k(h) - \widehat{\mathcal{D}}_k^+(h)\|_{TV}) \cdot T$. If $1 - T \leq 3\bar{\epsilon}_{g+1}/4$, $k \leftarrow k+1$ and go to Step 4. Otherwise, skip the interval from $x_{h+1}$ to $x_{h+k-1}$. Update $V$ and go to Step 1.

Figure 2: Approximately learning $\mathrm{III}$ under product distributions

The remaining problem is to identify the length of this interval, that is, to estimate the probability that an error happens if we skip an interval. Let $\mathcal{D}_{1:k}(h)$ be the distribution of $x[h+1, h+k]$ over all strings given $I_{V \sqsubseteq x} = h$ and $\mathcal{D}_{1:k}^+(h)$ be the corresponding distribution over all positive strings given $I_{V \sqsubseteq x} = h$. The probability that an error happens due to skipping the next $k$ elements is the total variation distance between $\mathcal{D}_{1:k}(h)$ and $\mathcal{D}_{1:k}^+(h)$. Thanks to the independence between the elements in a string, it can be proved that $\|\mathcal{D}_{1:k}(h) - \mathcal{D}_{1:k}^+(h)\|_{TV}$ can be estimated within polynomially bounded error.

Because the lengths of skipped intervals in cases with different $I_{V \sqsubseteq x}$ could be different, the algorithm branches the classification tree to determine the skipped interval according to the value of $I_{V \sqsubseteq x}$. The algorithm runs the procedure above recursively on each branch. Figure 2 demonstrates this skipping strategy of the algorithm, where parameter $C$ is the maximum allowed number of skipped intervals on each path. Notice that the algorithm might not recover the complete pattern string $U$. Instead the hypothesis pattern string returned by the algorithm for one classification path is a subsequence of $U$ with skipped intervals. We provide a toy example to explain the skipping logic. Let $n = 4$, $\Sigma = \{$a, b, c$\}$ and $U = $ 'ab'. Strings are drawn from a product distribution such that $x_1$, $x_2$ and $x_4$ are uniformly distributed over $\Sigma$ but $x_2$ is almost surely 'a'. The algorithm first estimates $\Pr[y = +1 \mid x_1 = a]$ for each $a \in \Sigma$ and finds the value of $x_1$ matters little to the positivity. It then estimates the distance between the distribution of $x_1 x_2$ over all positive strings and that over all strings and finds the two distributions are close. However, when it moves on to estimate the distance between the distribution of $x_1 x_2 x_3$ over all positive strings and that over all strings, it gets a nonnegligible total variation distance. Therefore, the skipped interval is $x_1 x_2$. The algorithm finally outputs the hypothesis pattern string '$\Sigma\Sigma$b' which means skipping the first two symbols and matching symbol 'b' in the rest of the string.

**Theorem 4** *Under product distributions over instance space $\mathcal{I} = \Sigma^n$, given $\Pr[x_i = a] \geq t > 0$ for $\forall 1 \leq i \leq n$ and $\forall a \in \Sigma$, the algorithm PAC classifies any string that skips $C = O(1)$ intervals during the classification procedure with $O(sn^{C+2})$ conditional statistical queries from STAT$(\mathrm{III}, \mathcal{D})$ at tolerance*

$$\tau = \min\left\{ \frac{\bar{\epsilon}_1^2}{384}, \frac{\bar{\epsilon}_1}{8sn} \right\}$$

*or with $O(sn^{C+2})$ statistical queries from STAT$(\mathrm{III}, \mathcal{D})$ at tolerance*

$$\bar{\tau} = (\epsilon' - 2\tau) \cdot \min\left\{ \frac{t\bar{\epsilon}_1^2}{384}, \frac{\bar{\epsilon}_1}{8sn} \right\}$$

*where $\bar{\epsilon}_1 = (\epsilon'/3^{C+2})^{2^C}$ and $\epsilon' = \epsilon/(2n^C)$.*

**Proof** For the sake of the legitimacy and feasibility of the algorithm, we make an initial query to estimate the probability $\Pr[I_{V \sqsubseteq x} = h]$ for each $h$ at tolerance $\tau$. Denote $\epsilon' = \epsilon/(2n^C)$. If the answer is $\leq \epsilon' - \tau$, then $\Pr[I_{V \sqsubseteq x} = h] \leq \epsilon'$ is negligible and we won't consider such cases because any of them happens with probability $\leq \epsilon/2$. Otherwise we have $\Pr[I_{V \sqsubseteq x} = h] \geq \epsilon' - 2\tau$. With the lower bound assumption that $\Pr[x_i = a] \geq t > 0$ for $\forall 1 \leq i \leq n$ and $\forall a \in \Sigma$, the legitimacy and feasibility are assured. Thus bounding the classification error in the nonnegligible cases within $\epsilon/2$ establishes a total error bound $\epsilon$. Because there are at most $n^C$ nonnegligible cases, the problem reduces to bounding the classification error for each within $\epsilon'$.

In the learning procedure, the algorithm skips an interval $x[i_1, i_2]$ given $I_{V \sqsubseteq x} = h$ based on the assumption that the interval $x[i_1, i_2]$ matches some segment next to $V$ in the pattern string $U$. Let $\iota_g$ be the indicator for the event that the assumption is false in the first $g$ skipped intervals and denote probability $\epsilon_g = \mathbf{E}\iota_g$. Let $\epsilon_0 = 0$. Note that $\epsilon_g$ serves as an upper bound for the probability of misclassification due to skipping the first $g$ intervals, because there are some lucky cases where the assumption doesn't hold but the algorithm still makes correct classifications. To ensure the accuracy of the algorithm, it suffices to prove $\epsilon_g$ is small. Let $\bar{\epsilon}_{g+1} = 8\sqrt{3\epsilon_g}$ for $g \geq 1$ and $\bar{\epsilon}_1$ as defined in the theorem. We will prove $\epsilon_{g+1} \leq \bar{\epsilon}_{g+1}$ so that by induction and taking the minimum tolerance among all $g \leq C$ we then have the overall tolerances $\tau$ and $\bar{\tau}$ as claimed in the statement.

Let $a_+, a'_+$ be two (not necessarily distinct) symbols in the next element of $U$ and $a_-, a'_-$ be two (not necessarily distinct) symbols not in the next element of $U$. We have $|P(+|a_+, h) - P(+|a'_+, h)| \leq \epsilon_g$ and likewise $|P(+|a_-, h) - P(+|a'_-, h)| \leq \epsilon_g$. Let $P_i(+|a, h) = P(+|a, h, \iota_g = i)$ and denote $\Delta = P(+|a_+, h) - P(+|a_-, h)$ and $\Delta_i = P_i(+|a_+, h) - P_i(+|a_-, h)$ for $i \in \{0, 1\}$. As a consequence, $\Delta = \epsilon_g \Delta_1 + (1 - \epsilon_g)\Delta_0$ and $\Delta_0 = \frac{\Delta - \epsilon_g \Delta_1}{1 - \epsilon_g} \geq \frac{\Delta - \epsilon_g}{1 - \epsilon_g}$. Therefore, $\Delta > \epsilon_g$ implies $\Delta_0 > 0$. In the other direction, $\Delta_0 = \frac{\Delta - \epsilon_g \Delta_1}{1 - \epsilon_g} \leq 2(\Delta + \epsilon_g)$.

For each $h$ we make a statistical query to estimate $P(+|a, h)$ for each $a \in \Sigma$ at tolerance $\tau_1 = \bar{\epsilon}_{g+1}^2/384$. If the minimum $\Delta$ among all pairs of $(a_+, a_-)$, denoted by $\Delta_{\min}$, is $> 6\tau_1$, the results of queries for $P(+|a, h)$ must form two distinguishable clusters, where the maximum difference inside one cluster is $\leq 4\tau_1$ while the minimum gap between two clusters is $> 4\tau_1$. According to Lemma 11, the set of symbols with larger query answers is the next element in $U$ because $\Delta > \epsilon_g$ holds for all pairs of $(a_+, a_-)$.

Otherwise, the difference $\Delta_0 \leq 2(\Delta_{\min} + 2\epsilon_g + \epsilon_g) \leq \bar{\epsilon}_{g+1}^2/16$ for all $h$. Let $x' = xz$ where $z$ is an infinite string under the uniform distribution. Let $E_h(1, i)$ be the event that matching the next element in $U$ consumes exactly $i$ symbols in string $x'$ given $I_{V \sqsubseteq x'} = h$ and $\iota_g = 0$. Define probability $R_h(1, i) = \Pr[E_h(1, i)]$. Let conditional probability $P_0(+|E_h(1, i))$ be the probability of positivity conditioned on event $E_h(1, i)$. For example, $P_0(+|a_+, h)$ is indeed $P_0(+|E_h(1, 1))$.

Denote by $P_0(+|h) = \Pr[y = +1 \mid I_{V \sqsubseteq x} = h \wedge \iota_g = 0]$. Because $P_0(+|h) \geq P_0(+|a_-, h)$, we have

$$P_0(+|a_+, h) - P_0(+|h) \leq P_0(+|a_+, h) - P_0(+|a_-, h) < \frac{\bar{\epsilon}_{g+1}^2}{16}$$

while

$$P_0(+|a_+, h) - P_0(+|h) = \sum_{i=1}^{+\infty} R_h(1, i) \cdot (P_0(+|E_h(1, 1)) - P_0(+|E_h(1, i)))$$

Notice that probability $P_0(+|E_h(1, i))$ is monotonically non-increasing with respect to $i$. Then there must exist an integer $k \in [1, +\infty]$ such that $P_0(+|E_h(1, 1)) - P_0(+|E_h(1, i)) \leq \bar{\epsilon}_{g+1}/4$ for $\forall i \leq k$ and $P_0(+|E_h(1, 1)) - P_0(+|E_h(1, i)) \geq \bar{\epsilon}_{g+1}/4$ for $\forall i > k$. This implies

$$\sum_{i \leq k} R_h(1, i) \left(P_0(+|E_h(1, 1)) - P_0(+|E_h(1, i))\right) + \sum_{i > k} R_h(1, i) \left(P_0(+|E_h(1, 1)) - P_0(+|E_h(1, i))\right)$$
$$< \frac{\bar{\epsilon}_{g+1}^2}{16}$$

and

$$\frac{\bar{\epsilon}_{g+1}}{4} \sum_{i > k} R_h(1, i) < \frac{\bar{\epsilon}_{g+1}^2}{16}$$

Then we have $\sum_{i>k} R_h(1,i) < \bar\epsilon_{g+1}/4$. This means the next element in $U$ almost surely shows up in this $k$-length interval. In addition, the difference $P_0(+|E_h(1,1)) - P_0(+|E_h(1,i)) \le \bar\epsilon_{g+1}/4$ for $\forall i \le k$ means whether the next element in $U$ first shows up at $x_{h+1}$ or $x_{h+k}$ has little effect on the probability of positivity. There are two cases where an error happens due to skipping the interval. The first case is that the next element in $U$ doesn't occur within the interval, whose probability is $\sum_{i>k} R_h(1,i)$. The second case is that after matching the next element in $U$ at $x_{h+i}$ for some $1 \le i < k$, the value of $x[h+i+1, h+k]$ flips the class of the string. This happens with probability $\le P_0(+|E_h(1,1)) - P_0(+|E_h(1,k))$. By union bound, the probability of the errors because of skipping the interval $x[h+1, h+k]$ is at most $\bar\epsilon_{g+1}/2$.

It is worth pointing out that $k$ is an integer from 1 to $+\infty$ because when $i = 1$ the difference $P_0(+|E_h(1,1)) - P_0(+|E_h(1,i))$ is $0 \le \bar\epsilon_{g+1}/4$ and surely $k \ge 1$. This means this interval is not empty and ensures the existence of the interval we want. On the other hand, the value $k$ can be positive infinity but this makes no difference because the algorithm will skip everything until the end of a string.

After showing the existence of such an interval, we need to determine $k$ and locate the interval. Let $\mathcal{D}_k(h)$ be the distribution of $x_{h+k}$ and $\mathcal{D}_{1:k}(h)$ be the distribution of the $x[h+1, h+k]$ over all strings, both conditioned on $I_{V \sqsubseteq x} = h$. Also, let $\mathcal{D}_k^+(h)$ and $\mathcal{D}_{1:k}^+(h)$ be the corresponding distributions over all positive strings. We use $\widehat{\cdot}$ as estimators for probabilities or distributions. The probability that an error happens due to skipping the next $k$ letters is the total variation distance between $\mathcal{D}_{1:k}(h)$ and $\mathcal{D}_{1:k}^+(h)$. Recall that the total variation distance between two distributions $\mu_1$ and $\mu_2$ is

$$\|\mu_1 - \mu_2\|_{TV} = \frac{1}{2}\|\mu_1 - \mu_2\|_1 = \min_{(Y,Z)} \Pr[Y \neq Z]$$

where $Y \sim \mu_1$ and $Z \sim \mu_2$ are random variables over $\mu_1$ and $\mu_2$ respectively. The minimum is taken over all joint distributions $(Y, Z)$ such that the marginal distributions are still $\mu_1$ and $\mu_2$, i.e., $Y \sim \mu_1$ and $Z \sim \mu_2$.

Now let $Y \sim \mathcal{D}_{1:k}(h)$ and $Z \sim \mathcal{D}_{1:k}^+(h)$ be random strings over $\mathcal{D}_{1:k}(h)$ and $\mathcal{D}_{1:k}^+(h)$ respectively. Then

$$\|\mathcal{D}_{1:k}(h) - \mathcal{D}_{1:k}^+(h)\|_{TV} = \min_{(Y,Z)} \Pr[Y \neq Z]$$
$$= 1 - \max_{(Y,Z)} \Pr[Y = Z]$$
$$= 1 - \max_{(Y,Z)} \prod_{i=1}^{k} \Pr[Y_i = Z_i]$$
$$= 1 - \prod_{i=1}^{k} \max_{(Y,Z)} \Pr[Y_i = Z_i]$$
$$= 1 - \prod_{i=1}^{k} \left(1 - \min_{(Y,Z)} \Pr[Y_i \neq Z_i]\right)$$
$$= 1 - \prod_{i=1}^{k} \left(1 - \|\mathcal{D}_i(h) - \mathcal{D}_i^+(h)\|_{TV}\right)$$

because of the independence between the symbols in a string and the fact that all minimums and maximums are taken over all joint distributions $(Y, Z)$ such that the marginal distributions are still product distributions.

Thus we could estimate the global total variation distance $\|\mathcal{D}_{1:k}(h) - \mathcal{D}_{1:k}^+(h)\|_{TV}$ through estimating the local variation distance $\|\mathcal{D}_i(h) - \mathcal{D}_i^+(h)\|_{TV}$ for each $1 \le i \le k$. Assume $\widehat{p}_1$ and $\widehat{p}_2$ are estimates of two probabilities $p_1$ and $p_2$ from a statistical query at some tolerance $\tau_0$. We have

$$|p_1 p_2 - \widehat{p}_1 \widehat{p}_2| = |p_1 p_2 - p_1 \widehat{p}_2 + p_1 \widehat{p}_2 - \widehat{p}_1 \widehat{p}_2|$$
$$= |p_1(p_2 - \widehat{p}_2) + (p_1 - \widehat{p}_1)\widehat{p}_2|$$
$$\le p_1 |p_2 - \widehat{p}_2| + |p_1 - \widehat{p}_1|\widehat{p}_2$$
$$\le (p_1 + \widehat{p}_2)\tau_0 \le 2\tau_0$$

By induction it can be proved that $\left|\prod_{i=1}^{k} p_i - \prod_{i=1}^{k} \widehat{p}_i\right| \leq k\tau_0$, which is a polynomial bound. For a probability $q$, let $q_i$ be the corresponding probability conditioned on $\iota_g = i$ for $i \in \{0, 1\}$. We have $q = \epsilon_g q_1 + (1 - \epsilon_g) q_0$ and

$$q_0 = \frac{q - \epsilon_g q_1}{1 - \epsilon_g} \geq q - \epsilon_g q_1 \geq q - \epsilon_g$$

In the other direction,

$$q_0 = \frac{q - \epsilon_g q_1}{1 - \epsilon_g} = \frac{q + \epsilon_g - \epsilon_g^2 - \epsilon_g q - \epsilon_g + \epsilon_g^2 + \epsilon_g q - \epsilon_g q_1}{1 - \epsilon_g}$$

$$= \frac{(q + \epsilon_g)(1 - \epsilon_g) - \epsilon_g(1 + q_1 - \epsilon_g - q)}{1 - \epsilon_g} \leq q + \epsilon_g$$

Note that here without loss of generality, we assume $\epsilon \leq \min\{(n-1)t, 24/(sn)\}$ so that $1 + q_1 - \epsilon_g - q \geq (n-1)t - \epsilon_g + q_1 > 0$ and $\epsilon_g \leq \bar{\epsilon}_{g+1}^2/192 < \bar{\epsilon}_{g+1}/(8sn)$. In PAC learning model a polynomial upper bound for error parameter $\epsilon$ is trivial. Because if a learning algorithm works with a small error bound, it automatically guarantees larger error bounds. As a consequence, $|q - q_0| \leq \epsilon_g$. In addition, using the definition of $\|\cdot\|_{TV}$,

$$
\begin{aligned}
\left| \|\mathcal{D}_i(h) - \mathcal{D}_i^+(h)\|_{TV} - \|\widehat{\mathcal{D}}_i(h) - \widehat{\mathcal{D}}_i^+(h)\|_{TV} \right| =& \frac{1}{2} \left| \|\mathcal{D}_i(h) - \mathcal{D}_i^+(h)\|_1 - \|\widehat{\mathcal{D}}_i(h) - \widehat{\mathcal{D}}_i^+(h)\|_1 \right| \\
\leq& \frac{1}{2} \left| \|\mathcal{D}_i(h) - \mathcal{D}_i^+(h) - \widehat{\mathcal{D}}_i(h) + \widehat{\mathcal{D}}_i^+(h)\|_1 \right| \\
\leq& \frac{1}{2} \left( \|\mathcal{D}_i(h) - \widehat{\mathcal{D}}_i(h)\|_1 + \|\mathcal{D}_i^+(h) - \widehat{\mathcal{D}}_i^+(h)\|_1 \right) \\
\leq& \frac{s}{2} \left( \|\mathcal{D}_i(h) - \widehat{\mathcal{D}}_i(h)\|_\infty + \|\mathcal{D}_i^+(h) - \widehat{\mathcal{D}}_i^+(h)\|_\infty \right)
\end{aligned}
$$

Hence, if we make statistical queries $\chi'_{V,a,h,i}$ and $\chi^+_{V,a,h,i}$ at tolerance $\tau_2 = \bar{\epsilon}_{g+1}/(8sn)$ and because $\bar{\epsilon}_{g+1}/(8sn) + \epsilon_g < \bar{\epsilon}_{g+1}/(4sn)$, the noise on $\|\mathcal{D}_i(h) - \mathcal{D}_i^+(h)\|_{TV}$ will be at most $\bar{\epsilon}_{g+1}/(4n)$ and we will be able to estimate $\|\mathcal{D}_{1:k}(h) - \mathcal{D}_{1:k}^+(h)\|_{TV}$ within error $k\bar{\epsilon}_{g+1}/(4n) \leq \bar{\epsilon}_{g+1}/4$. If $\|\widehat{\mathcal{D}}_{1:k}(h) - \widehat{\mathcal{D}}_{1:k}^+(h)\|_{TV} \geq 3\bar{\epsilon}_{g+1}/4$, then $\|\mathcal{D}_{1:k}(h) - \mathcal{D}_{1:k}^+(h)\|_{TV} \geq \bar{\epsilon}_{g+1}/2$. Otherwise, $\|\mathcal{D}_{1:k}(h) - \mathcal{D}_{1:k}^+(h)\|_{TV} < \bar{\epsilon}_{g+1}$ and we are still safe to increase $k$.

The algorithm does $O(sn^{C+2})$ queries $\chi_{V,a,h}$ at tolerance $\tau_1 = \bar{\epsilon}_{g+1}^2/384$, plus $O(sn^{C+2})$ queries $\chi'_{V,a,h,i}$ and $\chi^+_{V,a,h,i}$ at tolerance $\tau_2 = \bar{\epsilon}_{g+1}/(8sn)$. Thus by induction and taking the minimum tolerance among all $g \leq C$ we have the overall tolerances $\tau$ and $\bar{\tau}$ as claimed in the statement. ∎

## D  Proof and details from Section 5

### D.1  Proof of Lemma 4

Here we provide omitted proof and discussion of Lemma 4.

**Lemma 4 (in the main paper)** *Under general unrestricted string distributions, a concept class is PAC learnable over instance space $\Sigma^{\leq n}$ if and only if it is PAC learnable over instance space $\Sigma^n$.*

**Proof** *If direction.* Assume concept class $\mathcal{C}$ is PAC learnable from fixed-length strings with algorithm $\mathcal{A}$ under unrestricted general distributions. Because instance space $\Sigma^{\leq n} = \bigcup_{i \leq n} \Sigma^i$, we divide the sample $S$ into $n$ subsets $\{S_i\}$ where $S_i = \{x \mid |x| = i\}$. We make an initial statistical query to estimate probability $\Pr[|x| = i]$ for each $i \leq n$ at tolerance $\epsilon/(8n)$. We discard all $S_i$ with query answer $\leq 3\epsilon/(8n)$, because we know $\Pr[|x| = i] \leq \epsilon/(2n)$. There are at most $(n-1)$ such $S_i$ of low occurrence probabilities. The total probability that an instance comes from one of these ignored sets is at most $\epsilon/2$. Otherwise, $\Pr[|x| = i] \geq \epsilon/(4n)$ and we apply algorithm $\mathcal{A}$ on each $S_i$ with query answer $\geq 3\epsilon/(8n)$ with error parameter $\epsilon/2$. Because the probability of the condition is polynomially large, the algorithm is feasible. Finally, the error over the whole instance space will be bounded by $\epsilon$ and concept class $\mathcal{C}$ is PAC learnable over instance space $\Sigma^{\leq n}$.

```
Input: N labeled strings ⟨x^i, y^i⟩, string length n, alphabet Σ
Output: pattern string û
1.     û ← λ
2.     for ℓ ← 0 to n
3.         reward ← 1 × |Σ| all 0 vector
4.         for each a ∈ Σ
5.             for i ← 1 to N
6.                 if û ⊑ x^i
7.                     if y^i = +1
8.                         reward[a] ← reward[a] + (I_{û⊑x^i} − min{I_{ûa⊑x^i}, n + 1}) r_+
9.                     else
10.                        reward[a] ← reward[a] + (min{I_{ûa⊑x^i}, n + 1} − I_{û⊑x^i}) r_−
11.                    endif
12.                else
13.                    if y^i = +1
14.                        return û[1, ℓ − 1]
15.                    endif
16.                endif
17.            endfor
18.         endforeach
19.         û_{ℓ+1} ← argmax_{a∈Σ}{reward[a]}
20.         û ← û û_{ℓ+1}
21.     endfor
22.     return û
```

Figure 3: A greedy algorithm for learning ideal ⊔ from example oracle $EX(⊔, \mathcal{D})$

*Only-if direction.* This is an immediate consequence of the fact $\Sigma^n \subseteq \Sigma^{\leq n}$. ∎

Notice that Lemma 4 requires algorithm $\mathcal{A}$ to be applicable to any $S_i \mid i \leq n$. But this requirement can be weakened. There might not exist such a general algorithm $\mathcal{A}$. Instead we could have an algorithm $\mathcal{A}_i$ applicable to each subspace $S_i$ with non-negligible occurrence probability $\Pr[|x| = i] \geq \epsilon/(4n)$, then it is easy to see that Lemma 4 still holds in this case. Moreover, Lemma 4 makes no assumption on the string distribution. In the cases under restricted string distributions, here are two conditions that suffice to keep Lemma 4 hold: First, there is no assumption on the string length distribution; Second, we have an algorithm $\mathcal{A}_i$ applicable to instance space $S_i$ over marginal distribution $\mathcal{D}_{|x|=i}$ for each $1 \leq i \leq n$ such that $\Pr[|x| = i]$ is polynomially large.

## D.2 A heuristic greedy method

Figure 3 provides detailed pseudocode of the greedy method discussed in Section 5.

## D.3 Experiment settings and results

To make a comparison between the greedy method and kernel machines for empirical performance, we conducted a series of experiments in MATLAB on a workstation built with Intel i5-2500 3.30GHz CPU and 8GB memory. As discussed in Section 5, the running time of the kernel machine will be intolerable in practice when the sample size $N$ and the string length $n$ are large. Also, a pattern string $u$ of improper length will lead to a degenerate sample set which contains only positive or only negative example strings. To prevent this less interesting case from happening, we set $|u| = \lceil ns^{-1} \rceil$. Intuitively, the sample set will be evenly partitioned into two classes in expectation under the uniform distribution. However, in this case $n$ not being large demands the alphabet size $s$ not being large either.

Figure 4: Experiment results with NSF abstracts data set (training 1993; testing 1992)

Figure 5: Experiment results with NSF abstracts data set (training 1999; testing 1998)

Combining all these constraints together, the experiment settings are: alphabet size $s = 8$, size of training set = size of testing set = 1024. We vary the string length $n$ from 16 to 56 and let $|u| = \lceil ns^{-1} \rceil$. The pattern string $u$ is generated uniformly at random from $\Sigma^{|u|}$. Our tests are run on the NSF Research Award Abstracts data set [4]. We use the abstracts of year 1993 as the training set and those of year 1992 as the testing set. The tests are case-insensitive and all the characters except the subset from 'a(A)' to 'h(H)' are removed from the texts. The result texts are then partitioned into a set of strings of length $n$, which serve as the example strings. To be more robust against fluctuation from randomness, each test with a particular value of $n$ is run for 10 times and the medians of error rates and running times are taken as the final performance scores. Both lines climb as $n$ increases.

The experiment results are shown in Figure 4, with accuracy presented as line plot and efficiency demonstrated as bar chart. The overwhelming advantage of the greedy algorithm on efficiency is obvious. The kernel machine ran for hours in high dimensional cases, while the greedy method achieved even better accuracy within only milliseconds. The error rate of the greedy algorithm is always lower than that of the kernel machine as well.

It is worth noting that MATLAB started reporting no-convergence error of the kernel method when the string length $n$ reaches 56. Only successful runs of the kernel method were taken into account. Therefore, the performance of the kernel method when $n = 56$ is very unstable over some datasets. Figure 5 is an example where kernel method became unpredictable when no-convergence error happened. In this plot when $n = 56$ the kernel machine seems to have better accuracy than the greedy method, but considering that all the failed runs of the kernel machine were ruled out and only successful ones were taken into account, the apparent accuracy of the kernel method is shaky.