[Reviews · NeurIPS 2014]

Submitted by Assigned_Reviewer_42

The paper presents a number of positive and negative results on learning shuffle ideals,
generalizing and extending previous work on this topic.

Previously it was known that this class is not PAC learnable in general, but is
learnable under the uniform distributions. The extends the positive result
to several more popular distributions, and strengthens the negative result as well
establishing a hardness result for general distributions.

I have a very limited expertise in this area (hence I put a low confidence score),
but the results of the paper seem to be a sound, albeit limited in scope, contribution to it.
Summary: The paper presents a number of positive and negative results on learning shuffle ideals,
generalizing and extending previous work on this topic.

Submitted by Assigned_Reviewer_43

This paper investigates the PAC learnability of a sub-family of regular languages called the "shuffle ideals" – sets of strings containing at least one member of a fixed, finite set of strings as a (possibly non-contiguous) subsequence. It was previously known that these languages are not PAC learnable in general but are PAC learnable under the uniform distribution. This paper extends that PAC learnability result to other families of distributions (product, Markovian, and others). The results rely on showing that these languages are learnable in the statistical query model which in turns implies learnability in the classical PAC model.

Summary: This is a clearly written, well motivated paper and, although I am unfamiliar with work on language learning, it appears technically sound and of a high quality. The results, while incremental, do seem novel and push forward what is known about the PAC learnability of regular languages in specific cases.

Submitted by Assigned_Reviewer_44

This paper studies the problem of learning shffle ideals in the PAC-learning model under some simple distributions such as product distribution, or distributions generated by Markov chains. A shuffle ideal is a particularly simple type of regular language. Given alphabet \Sigma, and a string u =(u_1, …, u_L) the prime shuffle ideal generated by u is the set of strings obtained by inserting any strings s_0, …, s_L in u to get s_0 u_1 s_1 … u_L s_L. The shuffle ideal is a more general class of regular languages where instead of a string u we consider strings in a product set U = U_1 \times … \times U_L where U_i \subseteq \Sigma and take the union of the prime shuffle ideals generated by strings in U.

The problem of learning regular languages has many applications. The theoretical study of this problem is in the PAC model. This however seems to be a hard problem, and so people try to study special cases and shuffle ideals are one such case.

I think there’s a disconnect here between theory and practice as I don’t see any bridge between them mentioned in the paper under review or in [3]. However, I do find the problem being studied interesting in its own right; it seems natural and basic to me and thus could be of use in other contexts. The results are (seemingly) substantial generalization of previous work in that several of the results here are for learning shuffle ideals (as opposed to prime shuffle ideals) and hold in situations where the distribution is generated by a Markov chain, or distribution over \Sigma^n where each position from 1 to n has its own distribution and one independently samples a symbol from \Sigma for each position. The algorithms work in the statistical query model and thus also in the PAC-learning model (for specific distributions). The paper also gives a simple greedy heuristic for learning prime shuffle ideals in the general case without proving any guarantees. Experimental results show it to be superior to the previous work.

Thus the results seem to be interesting and non-trivial and represent progress. However, I have serious concerns regarding the technical parts as I now explain:
[The following is based on the first 9 pages.]

The writing is rather dry with little by way of proof outlines or intuitive explanation of the main ideas behind the proofs. The only explanation is after Theorem 3, which I didn't find very helpful.

There should be a comparison with [3] in terms of new techniques and ideas. A cursory glance at [3] reveals high-level similarities in the notation and algorithms. So I suppose the present paper is building upon [3] and that makes it important to include a comparison with [3].

The writing is not very clear; in most cases, where something is not clear I can guess what is meant but at some places I am not so sure. A few examples:
The algorithm in Figure 3 seems to be missing some initialization. In particular, what’s V? What does it mean to approximately learn a shuffle ideal? I couldn’t find it being defined anywhere. Presumably it means that the symmetric difference between the learned concept and the actual concept is small? And when one is learning a prime shuffle ideal what does it mean? Is it that the Hamming distance or edit distance between the learned string and the actual string generating the ideal is small? But then that would require some quantification of approximation and the Theorems 1, 2, 3 do not have it. What is \sigma in the statements of these theorems? Is it a parameter that one can choose or is it somehow related to approximation? Is L (as in Def. 2) a constant or logarithmic in n? I guess some small upper bound on L is needed to justify the use of conditioning like V \sqsubseteq x[1:n-1] in statistical query model. Also I’d expect L to show up in the statements of theorems, but it doesn’t.

Mention that the problem is information-theoretically solvable by VC-dimension arguments (i.e. polynomial number of samples suffice though the algorithm may not be efficient).

Thus I have to say that even the theorem statements are not completely clear to me. The paper would have been much better if the authors had devoted more space to one of the proofs (say Theorem 3), and given sufficient explanation to be convincing about its correctness. Other results could be stated in less detail to meet the page limit. This also makes me unable to comment on the correctness and novelty of the proofs, although I have no reason to have doubt.
Summary: Results are interesting but the exposition needs significant improvement. In particular, give clear statements of theorems; at least informally give the main idea behind the proof of at least one of the theorems (whichever you consider most important); also include a comparison with [3]. I think the paper is not acceptable in its current form.

EDIT: I have increased the score in view of further information that has become available.

Submitted by Assigned_Reviewer_45

This paper extends the previous state of the art results [3] in learning
shuffle ideals in two important directions. First, rather than learning
principal shuffle ideals, it learns ideals generated by augmented strings.
Second, the uniform distribution assumption in [3] is relaxed to include a
much broader class of distributions, including position-wise iid and some
Markov chains.

Although some of the proofs bear a superficial resemblance to [3], it is
clear that the authors introduce several novel ideas and techniques.
This vindicates the prediction in [3]: "It seems that genuinely new ideas
will be required to handle nonuniform distributions."
If I understood correctly, the authors were able to isolate unimodality as
the key property that allows for the efficient inference of the next
"augmented character". This simple and elegant property of distribution
also appears to be the tool that allowed for the learning of shuffle
ideals generated by augmented strings (as opposed to just principal ones).
[It might help the uninitiated reader if the authors
pointed out precisely where the techniques of [3] broke down and which
innovations were required to handle this more general setting.]

The extension to Markov chains is also of interest. One should note that
the minorization condition M(a,b) \ge c is fairly restrictive; for
example, it restricts the alphabet size to be \le 1/c.
Note, however, that minorization implies contraction:
the Doeblin contraction coefficient of the Markov chain is
upper-bounded by 1-sc, where s is the alphabet size
(see Lemma 2.2.2 in
http://www.cs.bgu.ac.il/~karyeh/thesis.pdf
). Hence, one can't help but wonder if the minorization condition
could be relaxed to Doeblin contraction, and if in fact unimodality
is again the key property that enables efficient inference.
This raises fascinating questions:
Are contracting Markov chains unimodal?
Can one prove a learnability result for all quantifiably unimodal distributions?

This being a solid theory paper, it would not have occurred to me to ask about experiments. It is therefore remarkable that actual --- encouraging --- experimental results are provided. It is also significant to note that the greedy heuristic significantly outperforms the baseline kernel methods, both accuracy-wise and speed-wise.

Very minor points:
- missing period in (44)
- presumably, the Statistical Queries in (9) and Thm 3 [1st display]
are "conditional"
- Missing Section # in Appendix line 453.
Summary: A very well-written, highly technical, highly innovative paper --- more than worthy
of publication in NIPS.
Author Feedback
Author rebuttal: Reviewer_42:
Thank you for the positive feedback and useful comments.

Reviewer_43:
Thank you for the positive feedback and useful comments.

Reviewer_44:
Thank you for your valuable comments.
1. Approximate learning a shuffle ideal: In this paper, approximate learning refers to PAC learning under the statistical query (SQ) model. As in Def.3, PAC learning means learning a concept from a sample drawn from a fixed unknown distribution such that with probability at least 1-\delta, the probability of disagreement between the output hypothesis and the actual concept on an example drawn from the same distribution is at most \epsilon. In SQ model, the sample is not accessible to the learner. Instead, oracle STAT is given, which takes as input a SQ {\chi,\tau}. Here \chi is a mapping from a labeled example to {0,1} and STAT returns an estimate for the expectation of \chi with additive error at most \tau.

Unlike the general example model, SQ model has tolerance parameter \tau and error parameter \epsilon, but no confidence parameter \delta. Oracle STAT is weaker than the example oracle EX, i.e., PAC learnability from STAT implies PAC learnability from EX, but not vice versa.

2. Quantification of approximation: The error parameter \epsilon is a reasonable quantification of approximation. However, \epsilon is usually given as an input of the algorithm. Instead, people evaluate a PAC learning algorithm by its efficiency with fixed accuracy requirement. In SQ model, the tolerance \tau and the number of SQ's needed by the algorithm are two important evaluation criteria of efficiency.

3. Learning a principal shuffle ideal: Presumably, prime shuffle ideal in the comments means principal shuffle ideal. As in Def.2, a principal shuffle ideal generated by a string u \in \Sigma^L is the regular language \Sigma^* u_1 \Sigma^* u_2 \Sigma^* ... \Sigma^* u_L \Sigma^*. Learning a principal shuffle ideal means learning a shuffle ideal given it is generated by a simple string.

4. Comparison with [3]: We omitted detailed comparison with [3] mainly because of the space limit. [3] is an important reference so we use similar notations and definitions for consistency with previous related works. The ideas of [3] do not work in the cases discussed in the present paper and only work under the uniform distribution. The factors in the query require exact and complete knowledge of the distribution and the concept class is also restricted to be principal shuffle ideals. Besides, the formula for quantity P(L,n,s) in [3] is exactly known but we don't have exact expressions of such quantities in more general setting. Our new algorithms are applicable to learning general shuffle ideals under non-uniform and unknown distributions with much weaker restrictions. New quantities are introduced and analyzed to demonstrate the efficiency and accuracy of the new algorithms.

5. Initialization of the algorithm in Section 5: We assume the initialization of all the SQ algorithms is the same, i.e., we start from V=\lambda. Sorry for the confusion. We will clarify this detail for every algorithm.

6. About L: L means the length of the pattern string U as defined in the paper, and we assume no knowledge of L. We don't need a particular upper bound on L. When L>n, the positive probability Pr[y=+1]=0. Then the algorithms will detect this and output a hypothesis classifying all examples as negative. Otherwise, L is upper-bounded by n. Thus we may treat L as an unknown parameter.

7. About \sigma: The \sigma in the paper should be s. This was a notation mistake when unifying the notations. We really apologize for this.

8. Proof outlines: Very helpful advice. We will add/improve some proof outlines to help readers better understand the main ideas without reading the appendix.

9. VC-dimension arguments: We will mention the VC-dimension arguments in the introduction.

Other comments will be taken into account in the revision.

Reviewer_45:
Thank you for the positive feedback and constructive comments.
1. Where the techniques of [3] broke down: Very good point. We omitted this point mainly because of the space limit. The SQ used in [3] has a factor 1/(s-1) to balance the probability contributions of \sigma=\bar{u}_{\ell+1} and \sigma \ne \bar{u}_{\ell+1} to the expected value of \chi_{u,a}. This requires the string distribution to be uniform and the shuffle ideal to be principal, making it not applicable to the more general setting. Also, because we have complete knowledge of the distribution, the formula for quantity P(L,n,s) in [3] is exactly known, but we don't have exact expressions of such quantities under non-uniform and unknown distributions. Our new algorithms are applicable to learning extended shuffle ideals under non-uniform and unknown distributions with much weaker restrictions. P(L,n,s) is no longer unimodal in non-uniform cases, so new quantities are introduced and analyzed to demonstrate the efficiency and accuracy of the algorithms.

2. Relaxation of the minorization condition: Thank you for the suggestion. This is a very interesting idea and we will follow it up.